# Multiple-Step Greedy Policies in Online and Approximate Reinforcement Learning

**Yonathan Efroni**[*]
jonathan.efroni@gmail.com

**Gal Dalal**[*]
gald@campus.technion.ac.il

**Bruno Scherrer**[†]
bruno.scherrer@inria.fr

**Shie Mannor**[*]
shie@ee.technion.ac.il

## Abstract

Multiple-step lookahead policies have demonstrated high empirical competence in Reinforcement Learning, via the use of Monte Carlo Tree Search or Model Predictive Control. In a recent work [5], multiple-step greedy policies and their use in vanilla Policy Iteration algorithms were proposed and analyzed. In this work, we study multiple-step greedy algorithms in more practical setups. We begin by highlighting a counter-intuitive difficulty, arising with soft-policy updates: even in the absence of approximations, and contrary to the 1-step-greedy case, monotonic policy improvement is not guaranteed unless the update stepsize is sufficiently large. Taking particular care about this difficulty, we formulate and analyze online and approximate algorithms that use such a multi-step greedy operator.

## 1   Introduction

The use of the 1-step policy improvement in Reinforcement Learning (RL) was theoretically investigated under several frameworks, e.g., Policy Iteration (PI) [18], approximate PI [2, 9, 13], and Actor-Critic [10]; its practical uses are abundant [22, 12, 25]. However, single-step based improvement is not necessarily the optimal choice. It was, in fact, empirically demonstrated that multiple-step greedy policies can perform conspicuously better. Notable examples arise from the integration of RL and Monte Carlo Tree Search [4, 28, 23, 3, 25, 24] or Model Predictive Control [15, 6, 27].

Recent work [5] provided guarantees on the performance of the multiple-step greedy policy and generalizations of it in PI. Here, we establish it in the two practical contexts of online and approximate PI. With this objective in mind, we begin by highlighting a specific difficulty: *softly updating* a policy with respect to (w.r.t.) a multiple-step greedy policy does not necessarily result in improvement of the policy (Section 4). We find this property intriguing since monotonic improvement is guaranteed in the case of soft updates w.r.t. the 1-step greedy policy, and is central to the analysis of many RL algorithms [10, 9, 22]. We thus engineer several algorithms to circumvent this difficulty and provide some non-trivial performance guarantees, that support the interest of using multi-step greedy operators. These algorithms assume access to a generative model (Section 5) or to an approximate multiple-step greedy policy (Section 6).

## 2   Preliminaries

Our framework is the infinite-horizon discounted Markov Decision Process (MDP). An MDP is defined as the 5-tuple $(\mathcal{S}, \mathcal{A}, P, R, \gamma)$ [18], where $\mathcal{S}$ is a finite state space, $\mathcal{A}$ is a finite action space,

---

[*]Department of Electrical Engineering, Technion, Israel Institute of Technology
[†]INRIA, Villers les Nancy, France

$P \equiv P(s'|s, a)$ is a transition kernel, $R \equiv r(s, a)$ is a reward function, and $\gamma \in (0, 1)$ is a discount factor. Let $\pi : \mathcal{S} \to \mathcal{P}(\mathcal{A})$ be a stationary policy, where $\mathcal{P}(\mathcal{A})$ is a probability distribution on $\mathcal{A}$. Let $v^\pi \in \mathbb{R}^{|\mathcal{S}|}$ be the value of a policy $\pi$, defined in state $s$ as $v^\pi(s) \equiv \mathbb{E}^\pi[\sum_{t=0}^\infty \gamma^t r(s_t, \pi(s_t))|s_0 = s]$. For brevity, we respectively denote the reward and value at time $t$ by $r_t \equiv r(s_t, \pi_t(s_t))$ and $v_t \equiv v(s_t)$. It is known that $v^\pi = \sum_{t=0}^\infty \gamma^t (P^\pi)^t r^\pi = (I - \gamma P^\pi)^{-1} r^\pi$, with the component-wise values $[P^\pi]_{s,s'} \triangleq P(s' \mid s, \pi(s))$ and $[r^\pi]_s \triangleq r(s, \pi(s))$. Lastly, let

$$q^\pi(s, a) = \mathbb{E}^\pi[\sum_{t=0}^\infty \gamma^t r(s_t, \pi(s_t)) \mid s_0 = s, a_0 = a]. \tag{1}$$

Our goal is to find a policy $\pi^*$ yielding the optimal value $v^*$ such that

$$v^* = \max_\pi (I - \gamma P^\pi)^{-1} r^\pi = (I - \gamma P^{\pi^*})^{-1} r^{\pi^*}. \tag{2}$$

This goal can be achieved using the three classical operators (equalities hold component-wise):

$$\forall v, \pi, \ T^\pi v = r^\pi + \gamma P^\pi v,$$
$$\forall v, \ Tv = \max_\pi T^\pi v,$$
$$\forall v, \ \mathcal{G}(v) = \{\pi : T^\pi v = Tv\},$$

where $T^\pi$ is a linear operator, $T$ is the optimal Bellman operator and both $T^\pi$ and $T$ are $\gamma$-contraction mappings w.r.t. the max norm. It is known that the unique fixed points of $T^\pi$ and $T$ are $v^\pi$ and $v^*$, respectively. The set $\mathcal{G}(v)$ is the standard set of 1-step greedy policies w.r.t. $v$.

## 3 The $h$- and $\kappa$-Greedy Policies

In this section, we bring forward necessary definitions and results on two classes of multiple-step greedy policies: $h$- and $\kappa$-greedy [5]. Let $h \in \mathbb{N}\setminus\{0\}$. The $h$-greedy policy $\pi_h$ outputs the first optimal action out of the sequence of actions solving a non-stationary, $h$-horizon control problem as follows:

$$\forall s \in \mathcal{S}, \ \pi_h(s) \in \arg\max_{\pi_0} \max_{\pi_1,..,\pi_{h-1}} \mathbb{E}^{\pi_0...\pi_{h-1}} \left[ \sum_{t=0}^{h-1} \gamma^t r(s_t, \pi_t(s_t)) + \gamma^h v(s_h) \mid s_0 = s \right].$$

Since the $h$-greedy policy can be represented as the 1-step greedy policy w.r.t. $T^{h-1}v$, the set of $h$-greedy policies w.r.t. $v$, $\mathcal{G}_h(v)$, can be formally defined as follows:

$$\forall v, \pi, \ T_h^\pi v = T^\pi T^{h-1} v,$$
$$\forall v, \ \mathcal{G}_h(v) = \{\pi : T_h^\pi v = T^h v\}.$$

Let $\kappa \in [0, 1]$. The set of $\kappa$-greedy policies w.r.t. a value function $v$, $\mathcal{G}_\kappa(v)$, is defined using the following operators:

$$\forall v, \pi, \ T_\kappa^\pi v = (I - \kappa\gamma P^\pi)^{-1}(r^\pi + (1 - \kappa)\gamma P^\pi v)$$
$$\forall v, \ T_\kappa v = \max_\pi T_\kappa^\pi v = \max_\pi (I - \kappa\gamma P^\pi)^{-1}(r^\pi + (1 - \kappa)\gamma P^\pi v) \tag{3}$$
$$\forall v, \ \mathcal{G}_\kappa(v) = \{\pi : T_\kappa^\pi v = T_\kappa v\}.$$

**Remark 1.** *A comparison of* (2) *and* (3) *reveals that finding the $\kappa$-greedy policy is equivalent to solving a $\kappa\gamma$-discounted MDP with shaped reward $r_{v,\kappa}^\pi \overset{def}{=} r^\pi + (1 - \kappa)\gamma P^\pi v$.*

In [5, Proposition 11], the $\kappa$-greedy policy was explained to be interpolating over all geometrically $\kappa$-weighted $h$-greedy policies. It was also shown that for $\kappa = 0$, the 1-step greedy policy is restored, while for $\kappa = 1$, the $\kappa$-greedy policy is the optimal policy.

Both $T_\kappa^\pi$ and $T_\kappa$ are $\xi_\kappa$ contraction mappings, where $\xi_\kappa = \frac{\gamma(1-\kappa)}{1-\gamma\kappa} \in [0, \gamma]$. Their respective fixed points are $v^\pi$ and $v^*$. For brevity, where there is no risk of confusion, we shall denote $\xi_\kappa$ by $\xi$. Moreover, in [5] it was shown that both the $h$- and $\kappa$-greedy policies w.r.t. $v^\pi$ are strictly better then $\pi$, unless $\pi = \pi^*$.

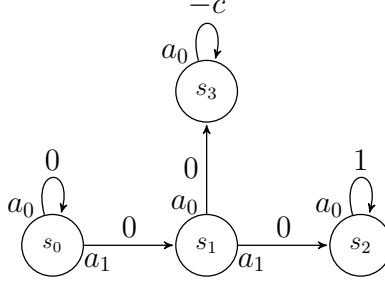

Figure 1: The Tightrope Walking MDP used in the counter example of Theorem 1.

Next, let

$$q_\kappa^\pi(s, a) = \max_{\pi'} \mathbb{E}^{\pi'}[\sum_{t=0}^{\infty} (\kappa\gamma)^t (r(s_t, \pi'(s_t)) + \gamma(1 - \kappa)v^\pi(s_{t+1}) \mid s_0 = s, a_0 = a]. \quad (4)$$

The latter is the optimal $q$-function of the surrogate, $\gamma\kappa$-discounted MDP with $v^\pi$-shaped reward (see Remark 1). Thus, we can obtain a $\kappa$-greedy policy, $\pi_\kappa \in \mathcal{G}_\kappa(v^\pi)$, directly from $q_\kappa^\pi$ :

$$\pi_\kappa(s) \in \arg\max_a q_\kappa^\pi(s, a), \ \forall s \in \mathcal{S}.$$

See that the greedy policy w.r.t. $q_{\kappa=0}^\pi(s, a)$ is the 1-step greedy policy since $q_{\kappa=0}^\pi(s, a) = q^\pi(s, a)$.

## 4 Multi-step Policy Improvement and Soft Updates

In this section, we focus on policy improvement of multiple-step greedy policies, performed with soft updates. Soft updates of the 1-step greedy policy have proved necessary and beneficial in prominent algorithms [10, 9, 22]. Here, we begin by describing an intrinsic difficulty in selecting the step-size parameter $\alpha \in (0, 1]$ when updating with multiple-step greedy policies. Specifically, denote by $\pi'$ such multiple-step greedy policy w.r.t. $v^\pi$. Then, $\pi_{\text{new}} = (1 - \alpha)\pi + \alpha\pi'$ is not necessarily better than $\pi$.

**Theorem 1.** *For any MDP, let $\pi$ be a policy and $v^\pi$ its value. Let $\pi_\kappa \in \mathcal{G}_\kappa(v^\pi)$ and $\pi_h \in \mathcal{G}_h(v^\pi)$ with $\kappa \in [0, 1]$ and $h > 1$. Consider the mixture policies with $\alpha \in (0, 1]$,*

$$\pi(\alpha, \kappa) \stackrel{def}{=} (1 - \alpha)\pi + \alpha\pi_\kappa,$$

$$\pi(\alpha, h) \stackrel{def}{=} (1 - \alpha)\pi + \alpha\pi_h.$$

*Then we have the following equivalences:*

1. *The inequality $v^{\pi(\alpha,\kappa)} \geq v^\pi$ holds for all MDPs if and only if $\alpha \in [\kappa, 1]$.*

2. *The inequality $v^{\pi(\alpha,h)} \geq v^\pi$ holds for all MDPs if and only if $\alpha = 1$.*

*The above inequalities hold entry-wise, with strict inequality in at least one entry unless $v^\pi = v^*$.*

*Proof sketch.* See Appendix A for the full proof. Here, we only provide a counterexample demonstrating the potential non-monotonicity of $\pi(\alpha, \kappa)$ when the stepsize $\alpha$ is not big enough. One can show the same for $\pi(\alpha, h)$ with the same example.

Consider the Tightrope Walking MDP in Fig. 1. It describes the act of walking on a rope: in the initial state $s_0$ the agent approaches the rope, in $s_1$ the walking attempt occurs, $s_2$ is the goal state and $s_3$ is repeatedly met if the agent falls from the rope, resulting in negative reward.

First, notice that by definition, $\forall v, \ \pi^* \in \mathcal{G}_{\kappa=1}(v)$. We call this policy the "confident" policy. Obviously, for any discount factor $\gamma \in (0, 1)$, $\pi^*(s_0) = a_1$ and $\pi^*(s_1) = a_1$. Instead, consider the "hesitant" policy $\pi_0(s) \equiv a_0 \ \forall s$. We now claim that for any $\alpha \in (0, 1)$ and

$$c > \frac{\alpha}{1 - \alpha} \quad (5)$$

the mixture policy, $\pi(\alpha, \kappa = 1) = (1 - \alpha)\pi_0 + \alpha\pi^*$, is not strictly better than $\pi_0$. To see this, notice that $v^{\pi_0}(s_1) < 0$ and $v^{\pi_0}(s_0) = 0$; i.e., the agent accumulates zero reward if she does not climb the rope. Thus, while $v^{\pi_0}(s_0) = 0$, taking any mixture of the confident and hesitant policies can result in $v^{\pi(\alpha, \kappa=1)}(s_0) < 0$, due to the portion of the transition to $s_1$ and its negative contribution. Based on this construction, let $\kappa \in [0, 1]$. To ensure $\pi^* \in \mathcal{G}_\kappa(v^\pi)$, we find it is necessary that

$$c \le \frac{\kappa}{1 - \kappa}. \tag{6}$$

To conclude, if both (5) and (6) are satisfied, the mixture policy does not improve over $\pi_0$. Due to the monotonicity of $\frac{x}{1-x}$, such a choice of $c$ is indeed possible for $\alpha < \kappa$. $\square$

Theorem 1 guarantees monotonic improvement for the 1-step greedy policy as a special case when $\kappa = 0$. Hence, we get that for any $\alpha \in (0, 1]$, the mixture of any policy $\pi$ and the 1-step greedy policy w.r.t. $v^\pi$ is monotonically better then $\pi$. To the best of our knowledge, this result was not explicitly stated anywhere. Instead, it appeared within proofs of several famous results, e.g, [10, Lemma 5.4], [9, Corollary 4.2], and [21, Theorem 1].

In the rest of the paper, we shall focus on the $\kappa$-greedy policy and extend it to the online and the approximate cases. The discovery that the $\kappa$-greedy policy w.r.t. $v^\pi$ is not necessarily strictly better than $\pi$ will guide us in appropriately devising algorithms.

## 5  Online $\kappa$-Policy Iteration with Cautious Soft Updates

In [5], it was shown that using the $\kappa$-greedy policy in the improvement stage leads to a convergent PI procedure – the $\kappa$-PI algorithm. This algorithm repeats i) finding the optimal policy of small-horizon surrogate MDP with shaped reward, and ii) calculating the value of the optimal policy and use it to shape the reward of next iteration. Here, we devise a practical version of $\kappa$-PI, which is model-free, online and runs in two timescales; i.e, it performs i) and ii) simultaneously.

The method is depicted in Algorithm 1. It is similar to the asynchronous PI analyzed in [16], except for two major differences. First, the fast timescale tracks both $q^\pi, q_\kappa^\pi$ and not just $q^\pi$. Thus, it enables access to *both* the 1-step-greedy and $\kappa$-greedy policies. The 1-step greedy policy is attained via the $q^\pi$ estimate, which is plugged into a $q$-learning [29] update rule for obtaining the $\kappa$-greedy policy. The latter essentially solves the surrogate $\kappa\gamma$-discounted MDP (see Remark 1). The second difference is in the slow timescale, in which the policy is updated using a new operator, $b_s$, as defined below. To better understand this operator, first notice that in Stochastic Approximation methods such as Algorithm 1, the policy is improved using soft updates with decaying stepsizes. However, as Theorem 1 states, monotonic improvement is not guaranteed below a certain stepsize value. Hence, for $q, q_\kappa \in \mathbb{R}^{|\mathcal{S} \times \mathcal{A}|}$ and policy $\pi$, we set $b_s(q, q_\kappa, \pi)$ to be the $\kappa$-greedy policy only when assured to have improvement:

$$b_s(q, q_\kappa, \pi) = \begin{cases} a_\kappa(s) & \text{if } q(s, a_\kappa) \ge v^\pi(s), \\ a_{\text{1-step}}(s) & \text{else,} \end{cases}$$

where $a_\kappa(s) \stackrel{\text{def}}{=} \arg\max_a q_\kappa(s, a)$, $a_{\text{1-step}}(s) \stackrel{\text{def}}{=} \arg\max_a q(s, a)$, and $v^\pi(s) = \sum_a \pi(a \mid s)q(s, a)$.

We respectively denote the state and state-action-pair visitation counters after the $n$-th time-step by $\nu_n(s) \stackrel{\text{def}}{=} \sum_{k=1}^n \mathbb{1}_{s=s_k}$ and $\phi_n(s, a) \stackrel{\text{def}}{=} \sum_{k=1}^n \mathbb{1}_{(s,a)=(s_k,a_k)}$. The stepsize sequences $\mu_f(\cdot), \mu_s(\cdot)$ satisfy the common assumption (B2) in [16], among which $\lim_{n\to\infty} \mu_s(n)/\mu_f(n) \to 0$. The second moments of $\{r_n\}$ are assumed to be bounded. Furthermore, let $\nu$ be some measure over the state space, s.t. $\forall s \in \mathcal{S}, \nu(s) > 0$. Then, we assume to have a generative model $\mathbb{G}(\nu, \pi)$, using which we sample state $s \sim \nu$, sample action $a \sim \pi(s)$, apply action $a$ and receive reward $r$ and next state $s'$.

The fast-timescale update rules in lines 6 and 8 can be jointly written as the sum of $H_\kappa^\pi(q, q_\kappa)$ (defined below) and a martingale difference noise.

---
**Algorithm 1** Two-Timescale Online $\kappa$-Policy-Iteration
---
1: **initialize:** $\pi_0, q_0, q_{\kappa,0}$.
2: **for** $n = 0, \dots$ **do**
3:     $s_n, a_n, r_n, s'_n \sim \mathbb{G}(\nu, \pi_n)$
4:     # Fast-timescale updates
5:     $\delta_n = r_n + \gamma v_n^\pi(s'_n) - q_n(s_n, a_n)$
6:     $q_{n+1}(s_n, a_n) \leftarrow q_n(s_n, a_n) + \mu_f(\phi_{n+1}(s_n, a_n))\delta_n$
7:     $\delta_{\kappa,n} = r_n + \gamma(1 - \kappa)v_n^\pi(s'_n) + \kappa\gamma \max_{a'} q_{\kappa,n}(s'_n, a') - q_{\kappa,n}(s_n, a_n)$
8:     $q_{\kappa,n+1}(s_n, a_n) \leftarrow q_{\kappa,n}(s_n, a_n) + \mu_f(\phi_{n+1}(s_n, a_n))\delta_{\kappa,n}$
9:     # Slow-timescale updates
10:    $\pi_{n+1}(s_n) \leftarrow \pi_n(s_n) + \mu_s(\nu_{n+1}(s_n))(b_{s_n}(q_{n+1}, q_{\kappa,n+1}, \pi_n) - \pi_n(s_n))$
11: **end for**
12: **return:** $\pi$
---

**Definition 1.** *Let $q, q_\kappa \in \mathbb{R}^{|\mathcal{S}||\mathcal{A}|}$. The mapping $H_\kappa^\pi : \mathbb{R}^{2|\mathcal{S}||\mathcal{A}|} \to \mathbb{R}^{2|\mathcal{S}||\mathcal{A}|}$ is defined as follows* $\forall(s, a) \in \mathcal{S} \times \mathcal{A}$.

$$H_\kappa^\pi(q, q_\kappa)(s, a) \overset{def}{=} \begin{bmatrix} r(s, a) + \gamma\mathbb{E}_{s', a^\pi}q(s', a^\pi) \\ r(s, a) + \gamma(1 - \kappa)\mathbb{E}_{s', a^\pi}q(s', a^\pi) + \kappa\gamma\mathbb{E}_{s'} \max_{a'} q_\kappa(s', a') \end{bmatrix},$$

*where $s' \sim P(\cdot \mid s, a), a^\pi \sim \pi(s')$.*

The following lemma shows that, given a fixed $\pi$, $H_\kappa^\pi$ is a contraction, equivalently to [16, Lemma 5.3] (see Appendix B for the proof).

**Lemma 2.** *$H_\kappa^\pi$ is a $\gamma$-contraction in the max-norm. Its fixed point is $[\, q^\pi, q_\kappa^\pi \,]^\top$, as defined in (1), (4).*

Finally, based on several intermediate results given in Appendix C and relaying on Lemma 2, we establish the convergence of Algorithm 1.

**Theorem 3.** *The coupled process $(q_n, q_{\kappa,n}, \pi_n)$ in Algorithm 1 converges to the limit $(q^*, q^*, \pi^*)$, where $q^*$ is the optimal q-function and $\pi^*$ is the optimal policy.*

For $\kappa = 1$, the fast-timescale update rule in line 8 corresponds to that of $q$-learning [29]. For that $\kappa$, Algorithm 1 uses an estimated optimal $q$-function to update the current policy when improvement is assured. For $\kappa < 1$, the estimated $\kappa$-dependent optimal $q$-function (see (4)) is used, again with the 'cautious' policy update. Moreover, Algorithm 1 combines an off-policy algorithm, i.e., $q$-learning, with an on-policy Actor-Critic algorithm. To the best of our knowledge, this is the first appearance of these two approaches combined in a single algorithm.

## 6 Approximate $\kappa$-Policy Iteration with Hard Updates

Theorem 1 establishes the conditions required for guaranteed monotonic improvement of softly-updated multiple-step greedy policies. The algorithm in Section 5 then accounts for these conditions to ensure convergence. Contrarily, in this section, we derive and study algorithms that perform hard policy-updates. Specifically, we generalize the prominent Approximate Policy Iteration (API) [13, 7, 11] and Policy Search by Dynamic Programming (PSDP) [1, 19]. For both, we obtain performance guarantees that exhibit a tradeoff in the choice of $\kappa$, with optimal performance bound achieved with $\kappa > 0$. That is, our approximate $\kappa$-generalized PI methods outperform the 1-step greedy approximate PI methods in terms of best known guarantees.

For the algorithms here we assume an oracle that returns a $\kappa$-greedy policy with some error. Formally, we denote by $\mathcal{G}_{\kappa,\delta,\nu}(v)$ the set of approximate $\kappa$-greedy policies w.r.t. $v$, with $\delta$ approximation error under some measure $\nu$.

**Definition 2** (Approximate $\kappa$-greedy policy). *Let $v : \mathcal{S} \to \mathbb{R}$ be a value function, $\delta \geq 0$ a real number and $\nu$ a distribution over $\mathcal{S}$. A policy $\pi \in \mathcal{G}_{\kappa,\delta,\nu}(v)$ if $\nu T_\kappa^\pi v \geq \nu T_\kappa v - \delta$.*

Such a device can be implemented using existing approximate methods, e.g., Conservative Policy Iteration (CPI) [9], approximate PI or VI [7], Policy Search [21], or by having an access to an approximate model of the environment. The approximate $\kappa$-greedy oracle assumed here is less

restrictive than the one assumed in [5]. There, a uniform error over states was assumed, whereas here, the error is defined w.r.t. a specific measure, $\nu$. For practical purposes, $\nu$ can be thought of as the initial sampling distribution to which the MDP is initialized. Lastly, notice that the larger $\kappa$ is, the harder it is to solve the surrogate $\kappa\gamma$-discounted MDP since the discount factor is bigger [17, 26, 8]; i.e., the computational cost of each call to the oracle increases.

Using the concept of *concentrability coefficients* introduced in [13] (there, they were originally termed "diffusion coefficients"), we follow the line of work in [13, 14, 7, 19, 11] to prove our performance bounds. This allows a direct comparison of the algorithms proposed here with previously studied approximate 1-step greedy algorithms. Namely, our bounds consist of concentrability coefficients $C^{(1)}, C^{(2)}, C^{(2,k)}$ and $C^{\pi^*(1)}$ from [19, 11], as well as two new coefficients $C_\kappa^{\pi^*}$ and $C_\kappa^{\pi^*(1)}$.

**Definition 3** (Concentrability coefficients [19, 11])**.** *Let $\mu, \nu$ be some measures over $\mathcal{S}$. Let $\{c(i)\}_{i=0}^\infty$ be the sequence of the smallest values in $[1, \infty) \cup \{\infty\}$ such that for every $i$, for all sequences of deterministic stationary policies $\pi_1, \pi_2, .., \pi_i$, $\mu \prod_{j=1}^{i} P^{\pi_j} \leq c(i)\nu$. Let $C^{(1)}(\mu, \nu) = (1 - \gamma) \sum_{i=0}^\infty \gamma^i c(i)$ and $C^{(2,k)}(\mu, \nu) = (1 - \gamma)^2 \sum_{i,j=0}^\infty \gamma^{i+j} c(i + j + k)$. For brevity, we denote $C^{(2,0)}(\mu, \nu)$ as $C^{(2)}(\mu, \nu)$. Similarly, let $\{c^{\pi^*}(i)\}_{i=0}^\infty$ be the sequence of the smallest values in $[1, \infty) \cup \{\infty\}$ such that for every $i$, $\mu \left(P^{\pi^*}\right)^i \leq c^{\pi^*}(i)\nu$. Let $C^{\pi^*(1)}(\mu, \nu) = (1 - \gamma) \sum_{i=0}^\infty \gamma^i c^{\pi^*}(i)$.*

We now introduce two new concentrability coefficients suitable for bounding the worst-case performance of PI algorithms with approximate $\kappa$-greedy policies.

**Definition 4** ($\kappa$-Concentrability coefficients)**.** *Let $C_\kappa^{\pi^*(1)}(\mu, \nu) = \frac{\xi}{\gamma} C^{\pi^*(1)}(\mu, \nu) + (1 - \xi)\kappa c(0)$. Also, let $C_\kappa^{\pi^*}(\mu, \nu) \in [1, \infty) \cup \{\infty\}$ be the smallest value s.t. $d_{\kappa,\mu}^{\pi^*} \leq C_\kappa^{\pi^*}(\mu, \nu)\nu$, where $d_{\kappa,\mu}^{\pi^*} = (1 - \xi)\mu(I - \xi D_\kappa^{\pi^*} P^{\pi^*})^{-1}$ is a probability measure and $D_\kappa^\pi = (1 - \kappa\gamma)(I - \kappa\gamma P^\pi)^{-1}$ is a stochastic matrix.*

In the definitions above, $\nu$ is the measure according to which the approximate improvement is guaranteed, while $\mu$ specifies the distribution on which one measures the loss $\mathbb{E}_{s \sim \mu}[v^*(s) - v^{\pi_k}(s)] = \mu(v^* - v^{\pi_k})$ that we wish to bound. From Definition 4 it holds that $C_{\kappa=0}^{\pi^*}(\mu, \nu) = C^{\pi^*}(\mu, \nu)$; the latter was previously defined in, e.g, [19, Definition 1].

Before giving our performance bounds, we first study the behavior of the coefficients appearing in them. The following lemma sheds light on the behavior of $C_\kappa^{\pi^*}(\mu, \nu)$. Specifically, it shows that under certain constructions, $C_\kappa^{\pi^*}(\mu, \nu)$ decreases[3] as $\kappa$ increases (see proof in Appendix D).

**Lemma 4.** *Let $\nu(\alpha) = (1 - \alpha)\nu + \alpha\mu$. Then, for all $\kappa' > \kappa$, there exists $\alpha^* \in (0, 1)$ such that $C_{\kappa'}^{\pi^*}(\mu, \nu(\alpha^*)) \leq C_\kappa^{\pi^*}(\mu, \nu)$. The inequality is strict for $C_\kappa^{\pi^*}(\mu, \nu) > 1$. For $\mu = \nu$ this implies that $C_\kappa^{\pi^*}(\nu, \nu)$ is a decreasing function of $\kappa$.*

Definition 4 introduces two coefficients with which we shall derive our bounds. Though traditional arithmetic relations between them do not exist, they do comply to some notion of ordering.

**Remark 2** (Order of concentrability coefficients)**.** *In [19], an order between the concentrability coefficients was introduced: a coefficient $A$ is said to be strictly better than $B$ — a relation we denote with $A \prec B$ — if and only if i) $B < \infty$ implies $A < \infty$ and ii) there exists an MDP for which $A < \infty$ and $B = \infty$. Particularly, it was argued that*

$$C^{\pi^*}(\mu, \nu) \prec C^{\pi^*(1)}(\mu, \nu) \prec C^{(1)}(\mu, \nu) \prec C^{(2)}(\mu, \nu), \text{ and}$$

$$C^{(2,k_1)}(\mu, \nu) \prec C^{(2,k_2)}(\mu, \nu) \text{ if } k_2 < k_1.$$

*In this sense, $C_\kappa^{\pi^*(1)}(\mu, \nu)$ is analogous to $C^{\pi^*(1)}(\mu, \nu)$, while its definition might suggest improvement as $\kappa$ increases. Moreover, combined with the fact that $C_\kappa^{\pi^*}(\mu, \nu)$ improves as $\kappa$ increases, as Lemma 4 suggests, $C_\kappa^{\pi^*}(\mu, \nu)$ is better than all previously defined concentrability coefficients.*

## 6.1  $\kappa$-Approximate Policy Iteration

A natural generalization of API [13, 19, 11] to the multiple-step greedy policy is $\kappa$-API, as given in Algorithm 2. In each of its iterations, the policy is updated to the approximate $\kappa$-greedy policy w.r.t. $v^{\pi_{k-1}}$; i.e, a policy from the set $\mathcal{G}_{\kappa,\delta,\nu}(v^{\pi_{k-1}})$.

| **Algorithm 2** $\kappa$-API | **Algorithm 3** $\kappa$-PSDP |
|---|---|
| **initialize** $\kappa \in [0,1], \nu, \delta, v^{\pi_0}$ | **initialize** $\kappa \in [0,1], \nu, \delta, v^{\pi_0}, \Pi = [\,]$ |
| $v \leftarrow v^{\pi_0}$ | $v \leftarrow v^{\pi_0}$ |
| **for** $k = 1, ..$ **do** | **for** $k = 1, ..$ **do** |
| $\quad \pi_k \leftarrow \mathcal{G}_{\kappa,\delta,\nu}(v)$ | $\quad \pi_k \leftarrow \mathcal{G}_{\kappa,\delta,\nu}(v)$ |
| $\quad v \leftarrow v^{\pi_k}$ | $\quad v \leftarrow T_\kappa^{\pi_k} v$ |
| **end for** | $\quad \Pi \leftarrow$ Append$(\Pi, \pi_k)$ |
| **return** $\pi$ | **end for** |
| | **return** $\Pi$ |

The following theorem gives a performance bound for $\kappa$-API (see proof in Appendix E), with

$$C_{\kappa-\mathrm{API}}(\mu,\nu) = (1-\kappa)^2 C^{(2)}(\mu,\nu) + (1-\gamma)\kappa \left( (1-\kappa)C^{(1)}(\mu,\nu) + (1-\gamma\kappa)C_\kappa^{\pi^*(1)}(\mu,\nu) \right),$$

$$C_{\kappa-\mathrm{API}}^{(k,1)}(\mu,\nu) = (1-\kappa\gamma)\left( \kappa(1-\kappa\gamma)C_\kappa^{\pi^*}(\mu,\nu) + (1-\kappa)^2 C^{(1)}(\mu,\nu)) \right),$$

$$C_{\kappa-\mathrm{API}}^{(k,2)}(\mu,\nu) = (1-\kappa)\kappa \left( (1-\gamma)C^{(1)}(\mu,\nu) + g(\kappa)(1-\kappa)\gamma^k C^{(2,k)}(\mu,\nu) \right),$$

where $g(\kappa)$ is a bounded function for $\kappa \in [0,1]$.

**Theorem 5.** *Let $\pi_k$ be the policy at the $k$-th iteration of $\kappa$-API and $\delta$ be the error as defined in Definition 2. Then*

$$\mu(v^* - v^{\pi_k}) \leq \frac{C_{\kappa-\mathrm{API}}(\mu,\nu)}{(1-\gamma)^2}\delta + \xi^k \frac{R_{\max}}{1-\gamma}.$$

*Also, let $k = \left\lceil \frac{\log \frac{R_{max}}{\delta(1-\gamma)}}{1-\xi} \right\rceil$. Then $\mu(v^* - v^{\pi_k}) \leq \frac{C_{\kappa-\mathrm{API}}^{(k,1)}(\mu,\nu)}{(1-\gamma)^2}\log\left(\frac{R_{\max}}{(1-\gamma)\delta}\right)\delta + \frac{C_{\kappa-\mathrm{API}}^{(k,2)}(\mu,\nu)}{(1-\gamma)^2}\delta + \delta.$*

For brevity, we now discuss the first part of the statement; the same insights are true for the second as well. The bound for the original API is restored for the 1-step greedy case of $\kappa = 0$, i.e, $\mu(v^* - v^{\pi_k}) \leq \frac{C^{(2)}(\mu,\nu)}{(1-\gamma)^2}\delta + \frac{\gamma^k R_{\max}}{1-\gamma}$ [19, 11]. As in the case of API, our bound consists of a fixed approximation error term and a geometrically decaying term. As for the other extreme, $\kappa = 1$, we first remind that in the non-approximate case, applying $T_{\kappa=1}$ amounts to solving the original $\gamma$-discounted MDP in a single step [5, Remark 4]. In the approximate setup we investigate here, this results in the vanishing of the second, geometrically decaying term, since $\xi = 0$ for $\kappa = 1$. We are then left with a single constant approximation error: $\mu(v^* - v^{\pi_k}) \leq c(0)\delta$. Notice that $c(0)$ is independent of $\pi^*$ (see Definition 3). It represents the mismatch between $\mu$ and $\nu$ [9].

Next, notice that, by definition (see Definition 3), $C^{(2)}(\mu,\nu) > (1-\gamma)^2 c(0)$; i.e., $\frac{C^{(2)}(\mu,\nu)}{(1-\gamma)^2}\delta > c(0)\delta$. Given the discussion above, we have that the $\kappa$-API performance bound is *strictly* smaller with $\kappa = 1$ than with $\kappa = 0$. Hence, the bound suggests that $\kappa$-API is strictly better than the original API for $\kappa = 1$. Since all expressions there are continuous, this behavior does not solely hold point-wise.

**Remark 3** (Performance tradeoff). *Naively, the above observation would lead to the choice of $\kappa = 1$. However, it is reasonable to assume that $\delta$, the error of the $\kappa$-greedy step, itself depends on $\kappa$, i.e, $\delta \equiv \delta(\kappa)$. The general form of such dependence is expected to be monotonically increasing: as the effective horizon of the surrogate $\kappa\gamma$-discounted MDP becomes larger, its solution is harder to obtain (see Remark 1). Thus, Theorem 5 reveals a performance tradeoff as a function of $\kappa$.*

## 6.2 $\kappa$-Policy Search by Dynamic Programming

We continue with generalizing another approximate PI method – PSDP [1, 19]. We name it $\kappa$-PSDP and introduce it in Algorithm 3. This algorithm updates the policy differently from $\kappa$-API. However, similarly to $\kappa$-API, it uses hard updates. We will show this algorithm exhibits better performance than any other previously analyzed approximate PI method [19].

The $\kappa$-PSDP algorithm, unlike $\kappa$-API, returns a *sequence of deterministic policies*, $\Pi$. Given this sequence, we build a single, non-stationary policy by successively running $N_k$ steps of $\Pi[k]$, followed

by $N_{k-1}$ steps of $\Pi[k-1]$, etc, where $\{N_i\}_{i=1}^k$ are i.i.d. geometric random variables with parameter $1 - \kappa$. Once this process reaches $\pi_0$, it runs $\pi_0$ indefinitely. We shall refer to this non-stationary policy as $\sigma_{\kappa,k}$. Its value $v^{\sigma_{\kappa,k}}$ can be seen to satisfy

$$v^{\sigma_{\kappa,k}} = T_\kappa^{\Pi[k]} T_\kappa^{\Pi[k-1]} \cdots T_\kappa^{\Pi[1]} v^{\pi_0}.$$

This algorithm follows PSDP from [19]. Differently from it, the 1-step improvement is generalized to the $\kappa$-greedy improvement and the non-stationary policy behaves randomly. The following theorem gives a performance bound for it (see proof in Appendix F).

**Theorem 6.** *Let $\sigma_{\kappa,k}$ be the policy at the $k$-th iteration of $\kappa$-PSDP and $\delta$ be the error as defined in Definition 2. Then*

$$\mu(v^* - v^{\sigma_{\kappa,k}}) \leq \frac{C_\kappa^{\pi^*(1)}(\mu, \nu)}{1 - \xi} \delta + \xi^k \frac{R_{\max}}{1 - \gamma}.$$

*Also, let $k = \left\lceil \frac{\log \frac{R_{max}}{\delta(1-\gamma)}}{1-\xi} \right\rceil$. Then $\mu(v^* - v^{\sigma_{\kappa,k}}) \leq \frac{C_\kappa^{\pi^*}(\mu,\nu)}{(1-\xi)^2} \log\left(\frac{R_{\max}}{(1-\gamma)\delta}\right) \delta + \delta$.*

Compared to $\kappa$-API from the previous section, the $\kappa$-PSDP bound consists of a different fixed approximation error and a shared geometrically decaying term. Regarding the former, notice that $C_\kappa^{\pi^*(1)}(\mu, \nu) \prec C_{\kappa-\text{API}}(\mu, \nu)$, using the notation from Remark 2. This suggests that $\kappa$-PSDP is strictly better than $\kappa$-API in the metrics we consider, and is in line with the comparison of the original API to the original PSDP given in [19].

Similarly to the previous section, we again see that substituting $\kappa = 1$ gives a tighter bound than $\kappa = 0$. The reason is that $\frac{C_\kappa^{\pi^*(1)}(\mu,\nu)}{1-\gamma} \delta > c(0)\delta$, by definition (see Definition 3); i.e., we have that $\kappa$-PSDP is generally better than PSDP. Also, contrarily to $\kappa$-API, here we directly see the performance improvement as $\kappa$ increases due to the decrease of $C_\kappa^{\pi^*}$ prescribed in Lemma 4, for the construction given there. Moreover, the $\kappa$ tradeoff discussion in Remark 3 applies here as well.

An additional advantage of this new algorithm over PSDP is reduced space complexity. This can be seen from the $1 - \xi$ in the denominator in the choice of $k$ in the second part of Theorem 6. It shows that, since $\xi$ is a strictly decreasing function of $\kappa$, better performance is guaranteed with significantly fewer iterations by increasing $\kappa$. Since the size of stored policy $\Pi$ is linearly dependent on the number of iterations, larger $\kappa$ improves space efficiency.

# 7   Discussion and Future Work

In this work, we introduced and analyzed online and approximate PI methods, generalized to the $\kappa$-greedy policy, an instance of a multiple-step greedy policy. Doing so, we discovered two intriguing properties compared to the well-studied 1-step greedy policy, which we believe can be impactful in designing state-of-the-art algorithms. First, successive application of multiple-step greedy policies with a soft, stepsize-based update does not guarantee improvement; see Theorem 1. To mitigate this caveat, we designed an online PI algorithm with a 'cautious' improvement operator; see Section 5.

The second property we find intriguing stemmed from analyzing $\kappa$ generalizations of known approximate hard-update PI methods. In Section 6, we revealed a performance tradeoff in $\kappa$, which can be interpreted as a tradeoff between short-horizon bootstrap bias and long-rollout variance. This corresponds to the known $\lambda$ tradeoff in the famous TD($\lambda$).

The two characteristics above lead to new compelling questions. The first regards improvement operators: would a non-monotonically improving PI scheme necessarily not converge to the optimal policy? Our attempts to generalize existing proof techniques to show convergence in such cases have fallen behind. Specifically, in the online case, Lemma 5.4 in [10] does not hold with multiple-step greedy policies. Similar issues arise when trying to form a $\kappa$-CPI algorithm via, e.g., an attempt to generalize Corollary 4.2 in [9]. Another research question regards the choice of the parameter $\kappa$ given the tradeoff it poses. One possible direction for answering it could be investigating the concentrability coefficients further and attempting to characterize them for specific MDPs, either theoretically or via estimation. Lastly, a next indisputable step would be to empirically evaluate implementations of the algorithms presented here.

## Acknowledgments

This work was partially funded by the Israel Science Foundation under contract 1380/16.

## Footnotes

[3] A smaller coefficient is obviously better. The best value for any concentrability coefficient is 1.

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
