[Supplementary Material]

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

# A Proof of Theorem 1

We start with a generalization of a useful lemma; its original version appeared in, e.g., [20, Lemma 10].

**Lemma 7.** *Let $v$ be a value function, $\pi$ a policy, and $\kappa \in [0,1]$. Then*

$$T^\pi_\kappa v - v = (I - \kappa\gamma P^\pi)^{-1}(T^\pi v - v).$$

*Proof.* The proof is a straightforward generalization of the proof in [20, Lemma 10], and [9, Remark 6.1].

$$
\begin{aligned}
T^\pi_\kappa v - v &= (I - \kappa\gamma P^\pi)^{-1}(r^\pi + (1-\kappa)\gamma P^\pi v) - v \\
&= (I - \kappa\gamma P^\pi)^{-1}(r^\pi + (1-\kappa)\gamma P^\pi v - (I - \kappa\gamma P^\pi)v) \\
&= (I - \kappa\gamma P^\pi)^{-1}(r^\pi + \gamma P^\pi v - v) \\
&= (I - \kappa\gamma P^\pi)^{-1}(T^\pi v - v).
\end{aligned}
$$

$\square$

This elementary lemma relates the '$\kappa$-advantage' to the 1-step advantage and is useful to prove Theorem 1 and some following results.

First, since $\pi(\alpha, \kappa) = (1-\alpha)\pi + \alpha\pi_\kappa$, we have that

$$P^{\pi(\alpha,\kappa)} = (1-\alpha)P^\pi + \alpha P^{\pi_\kappa},$$
$$r^{\pi(\alpha,\kappa)} = (1-\alpha)r^\pi + \alpha r^{\pi_\kappa};$$

thus, since $v^\pi$ is the fixed-point of $T^\pi$,

$$T^{\pi(\alpha,\kappa)}v^\pi = (1-\alpha)T^\pi v^\pi + \alpha T^{\pi_\kappa}v^\pi = (1-\alpha)v^\pi + \alpha T^{\pi_\kappa}v^\pi. \tag{7}$$

Using this, we now prove the first statement of Theorem 1.

$$
\begin{aligned}
v^{\pi(\alpha,\kappa)} - v^\pi &= (I - \gamma P^{\pi(\alpha,\kappa)})^{-1}(T^{\pi(\alpha,\kappa)}v^\pi - v^\pi) \\
&= \alpha(I - \gamma P^{\pi(\alpha,\kappa)})^{-1}(T^{\pi_\kappa}v^\pi - v^\pi) \\
&= \alpha(I - \gamma P^{\pi(\alpha,\kappa)})^{-1}(I - \kappa\gamma P^{\pi_\kappa})(I - \kappa\gamma P^{\pi_\kappa})^{-1}(T^{\pi_\kappa}v^\pi - v^\pi) \\
&= \alpha(I - \gamma P^{\pi(\alpha,\kappa)})^{-1}(I - \kappa\gamma P^{\pi_\kappa})(T^{\pi_\kappa}_\kappa v^\pi - v^\pi) \\
&= \alpha(I - \gamma P^{\pi(\alpha,\kappa)})^{-1}(I - \gamma P^{\pi(\alpha,\kappa)} + \gamma(P^{\pi(\alpha,\kappa)} - \kappa P^{\pi_\kappa}))(T^{\pi_\kappa}_\kappa v^\pi - v^\pi) \\
&= \alpha\left(I + \gamma(I - \gamma P^{\pi(\alpha,\kappa)})^{-1}\right)((1-\alpha)P^\pi + (\alpha - \kappa)P^{\pi_\kappa})(T^{\pi_\kappa}_\kappa v^\pi - v^\pi). \tag{8}
\end{aligned}
$$

For the first relation we use Lemma 7 with $\kappa = 1$ and the fact that, by definition, $T^{\pi(\alpha,\kappa)}_{\kappa=1}v^{\pi(\alpha,\kappa)} = v^{\pi(\alpha,\kappa)}$. For the second relation we use (7), for the fourth we again use Lemma 7, and for the last relation we use that $P^{\pi(\alpha,\kappa)} - \kappa P^{\pi_\kappa} = (1-\alpha)P^\pi + (\alpha - \kappa)P^{\pi_\kappa}$.

Next, we show that for $\alpha \geq \kappa$, all terms in (8) are component-wise bigger than or equal to zero. First, using a Taylor expansion, $(I - \gamma P^{\pi(\alpha,\kappa)})^{-1} = \sum_t \gamma^t (P^{\pi(\alpha,\kappa)})^t \geq 0$ component-wise, since it is a weighted sum of transition matrices with positive weights. The same applies for $(1-\alpha)P^\pi + (\alpha - \kappa)P^{\pi_\kappa}$, when $\alpha \geq \kappa$. Thus, for $\alpha \geq \kappa$, $(I + \gamma(I - \gamma P^{\pi(\alpha,\kappa)})^{-1})((1-\alpha)P^\pi + (\alpha - \kappa)P^{\pi_\kappa}) \geq 0$ component-wise. Lastly, since $\pi_\kappa \in \mathcal{G}_\kappa(v^\pi)$, $v^\pi = T^\pi_\kappa v^\pi \leq T_\kappa v^\pi = T^{\pi_\kappa}_\kappa v^\pi$, with equality holding if and only if $v^\pi = v^*$ [5, Lemma 3]. Thus, $T^{\pi_\kappa}_\kappa v^\pi - v^\pi \geq 0$. This concludes the proof for the first statement, for the $\kappa$-greedy policy.

For the $\kappa$-greedy policy part of the proof for the second statement, we now provide more details on the counterexample presented in Section 4. For convenience, we bring the MDP example here again in Fig. 2. Consider the mixture of the "hesitant" and "confident" policies: $\pi(\alpha, \kappa = 1) = (1-\alpha)\pi_0 + \alpha\pi(\alpha, \kappa = 1)$. It can be shown that its value is

$$v^{\pi(\alpha,\kappa=1)}(s_0) = \frac{\gamma\alpha}{1 - \gamma(1-\alpha)}v^{\pi(\alpha,\kappa=1)}(s_1),$$

$$v^{\pi(\alpha,\kappa=1)}(s_1) = \gamma\frac{-c(1-\alpha) + \alpha}{1 - \gamma}.$$

Figure 2: The Tightrope Walking MDP used in the proof of Theorem 1. This class of MDPs is parametrized by $c > 0$.

Thus, we deduce that for any $\alpha \in (0, 1)$ and

$$c > \frac{\alpha}{1 - \alpha}, \tag{9}$$

$v^{\pi(\alpha, \kappa=1)}(s_0) < v^{\pi}(s_0) = 0$, i.e, the mixture policy, $\pi(\alpha, \kappa = 1)$, is not strictly better then $\pi_0$.

We now find the conditions to ensure that the $\kappa$-greedy policy w.r.t. $v^{\pi_0}$ is the optimal policy; this will generalize the above construction, made for $\kappa = 1$, to any $\kappa \in [0, 1]$. Observe that for any $c > 0$ and $\kappa$ it holds that $\pi_\kappa(s_1) = a_1 = \pi^*(s_1)$, where $\pi_\kappa \in \mathcal{G}_\kappa(v^{\pi_0})$. Thus, we solely need to consider the policy which is different than $\pi^*$ at state $s_0$, $\tilde{\pi}(s_0) = a_0 \neq \pi^*(s_0)$ and $\tilde{\pi}(s_1) = \pi^*(s_1)$. To find which condition ensures the $\kappa$-greedy policy w.r.t. $v^{\pi_0}$ is $\pi^*$ (and not $\tilde{\pi}$), we require

$$T_\kappa^{\pi^*} v^{\pi_0}(s_0) \geq T_\kappa^{\tilde{\pi}} v^{\pi_0}(s_0). \tag{10}$$

Satisfying this condition insures that $\pi^* \in \mathcal{G}_\kappa(v^{\pi_0})$. By definition,

$$T_\kappa^{\pi^*} v^{\pi_0}(s_0) = \mathbb{E}^{\pi^*} \left[ \sum_t (\kappa\gamma)^t (r(s_t, \pi^*(s_t)) + \gamma(1 - \kappa)v^{\pi_0}(s_{t+1}) \mid s_{t=0} = s_0 \right]$$

$$= (\kappa\gamma)^0 \left( \gamma(1 - \kappa)v^{\pi_0}(s_1) \right) + (\kappa\gamma)^1 \left( \gamma(1 - \kappa)v^{\pi_0}(s_2) \right) + \sum_{t=2}^{\infty} (\kappa\gamma)^t (1 + v^{\pi_0}(s_2))$$

$$= (\kappa\gamma)^0 \left( \gamma(1 - \kappa)(-\frac{\gamma c}{1 - \gamma}) \right) + (\kappa\gamma)^1 \left( \gamma(1 - \kappa)\frac{1}{1 - \gamma} \right) + \sum_{t=2}^{\infty} (\kappa\gamma)^t (1 + \gamma(1 - \kappa)\frac{1}{1 - \gamma})$$

$$= \gamma(1 - \kappa)(-\frac{\gamma c}{1 - \gamma}) + \kappa\gamma\frac{\gamma}{1 - \gamma}. \tag{11}$$

Similarly, and since $\tilde{\pi}(s_0) = a_0$, we have that

$$T_\kappa^{\tilde{\pi}} v^{\pi_0}(s_0) = 0 \tag{12}$$

Plugging (11) and (12) into (10), we get the condition

$$c \leq \frac{\kappa}{1 - \kappa}. \tag{13}$$

To finalize the counterexample and show that strict policy improvement is not guaranteed, we choose $c$ such that both (9) and (13) are satisfied. Such feasible choice exists when $\alpha < \kappa$, due to the monotonicity of $\frac{x}{1-x}$.

The monotonic improvement of $\pi(\alpha, h)$ for $\alpha = 1$ was proved in [5, Lemma 1]. To build the counter example, again consider the Tightrope MDP. Let $\pi_0$ be the 'hesitant' policy. For any $\gamma \in (0, 1)$, $h > 1$, it holds that $\pi^* \in \mathcal{G}_h(v^{\pi_0})$. Thus, it suffices to satisfy (9) alone to show that $\pi(\alpha, h) = (1 - \alpha)\pi_0 + \alpha\pi^*$ is not monotonically better then $\pi$. Large enough $c$ value ensures that.

# B  Proof of Lemma 2

We start by showing the contraction property of $H_\kappa^\pi$. Let $(s,a)$ be a fixed state-action pair, $Q_1, Q_2 \in \mathbb{R}^{2|\mathcal{S} \times \mathcal{A}|}$. For any state-action pair $(s,a)$, $Q_i(s,a)$ is a two-component vector. We denote its first component by $q_i(s,a)$ and its second component by $q_{i,\kappa}(s,a)$. See that

$$||q_1 - q_2||_\infty \le ||Q_1 - Q_2||_\infty, \tag{14}$$

$$||q_{1,\kappa} - q_{2,\kappa}||_\infty \le ||Q_1 - Q_2||_\infty. \tag{15}$$

Taking a component-wise absolute value, we have that

$$
\begin{aligned}
&|H_\kappa^\pi Q_1 - H_\kappa^\pi Q_2|(s,a)\\
=&|H_\kappa^\pi(q_1, q_{1,\kappa}) - H_\kappa^\pi(q_2, q_{2,\kappa})|(s,a)\\
=&\gamma \left[ \begin{matrix} |\mathbb{E}_{s',a^\pi}\left[ q_1(s',a^\pi)) - q_2(s', \pi(s'))\right]| \\ |(1-\kappa)\mathbb{E}_{s',a^\pi}\left[ q_1(s',a^\pi) - q_2(s',a^\pi))\right] + \kappa\mathbb{E}_{s'}[\max_{a'} q_{1,\kappa}(s',a') - \max_{a'} q_{2,\kappa}(s',a')]| \end{matrix} \right],
\end{aligned}
$$

where $s' \sim P(\cdot \mid s,a), a^\pi \sim \pi(s')$.

Let us focus on the first component of the above vector. We have that

$$\gamma|\mathbb{E}_{s',a^\pi}\left[ q_1(s',a^\pi) - q_2(s',a^\pi)\right]| \le \gamma||q_1 - q_2||_\infty \le \gamma||Q_1 - Q_2||_\infty,$$

where we used the standard bound, $|\mathbb{E}[X]| \le ||X||_\infty$ and (14). Similarly, for the second component, we have that

$$\gamma\left| \left( (1-\kappa)\mathbb{E}_{s',a^\pi}\left[ q_1(s',a^\pi) - q_2(s',a^\pi)\right] + \kappa\mathbb{E}_{s',a}[\max_{a'} q_{1,\kappa}(s',a') - \max_{a'} q_{2,\kappa}(s',a')]\right)\right|$$

$$\le\gamma\left( (1-\kappa)|\mathbb{E}_{s',a^\pi}\left[ q_1(s',a^\pi) - q_2(s',a^\pi)\right]| + \kappa\mathbb{E}_{s',a}[|\max_{a'} q_{1,\kappa}(s',a') - \max_{a'} q_{2,\kappa}(s',a')|]\right)$$

$$\le\gamma\left( (1-\kappa)|\mathbb{E}_{s',a^\pi}\left[ q_1(s',a^\pi) - q_2(s',a^\pi)\right]| + \kappa\mathbb{E}_{s',a'}[\max_{a'} |q_{1,\kappa}(s',a') - q_{2,\kappa}(s',a')|]\right)$$

$$\le\gamma\left( (1-\kappa)||q_1 - q_2||_\infty + \kappa||q_{1,\kappa} - q_{2,\kappa}||_\infty\right)$$

$$\le\gamma\left( (1-\kappa)||Q_1 - Q_2||_\infty + \kappa||Q_1 - Q_2||_\infty\right) = \gamma||Q_1 - Q_2||_\infty,$$

where for the first relation we used the triangle inequality, for the second we used the standard bound $|\max_{x \in \mathcal{X}} f(x) - \max_{x \in \mathcal{X}} g(x)| \le \max_{x \in \mathcal{X}} |f(x) - g(x)|$, for the third we used the bound $|\mathbb{E}[X]| \le ||X||_\infty$, and for the last (14)-(15).

From the above we get that

$$||H_\kappa^\pi Q_1 - H_\kappa^\pi Q_2||_\infty \le \gamma||Q_1 - Q_2||_\infty;$$

i.e., the operator $H_\kappa^\pi$ is a $\gamma$ contraction mapping in the max-norm.

It is clear that the fixed point of the first component is $q^\pi$. The fixed point of the second component is the fixed point of the optimal Bellman operator of the $\kappa\gamma$-discounted, reward shaped, surrogate MDP (see Remark 1). Its solution is, by construction, $q_\kappa^\pi$ (see (4)).

# C  Proof of Theorem 3

The proof of Theorem 3 follows the proof in [16, Section 5.1], with several generalizations given below.

## C.1  Lipschitzness of the Slow Time Scale Fixed-Point

Before following the main lemmas in [16] and showing they hold for Online $\kappa$-PI (Algorithm 1), we shall show that the solution of the fast-time scale ODE (found using a fixed-point argument), $[q^\pi, q_\kappa^\pi]$, is Lipschitz-continuous in the slow time-scale iterate, $\pi$.

**Lemma 8.** *Let $\pi : \mathcal{S} \times \mathcal{A} \to [0,1]$ be a stochastic policy. For any $\pi_1, \pi_2$ and $q_1, q_2 \in \mathbb{R}^{|\mathcal{S} \times \mathcal{A}|}$, let*

$$||\pi_1 - \pi_2||_\infty \stackrel{def}{=} \max_s \sum_a |\pi_1(a \mid s) - \pi_2(a \mid s)|,$$

$$||q_1 - q_2||_\infty \stackrel{def}{=} \max_{s,a} |q_1(s,a) - q_2(s,a)|.$$

*Then $q^\pi$ and $q^\pi_\kappa$ are Lipschitz-continuous in $\pi$ in the max-norm; i.e.,*

$$||q^{\pi_1} - q^{\pi_2}||_\infty \le L_a ||\pi_1 - \pi_2||_\infty,$$
$$||q^{\pi_1}_\kappa - q^{\pi_2}_\kappa||_\infty \le L_b ||\pi_1 - \pi_2||_\infty,$$

*where $L_a, L_b > 0$, are functions of $\gamma, \kappa, R_{\max}$.*

*Proof.* We start by proving that $||v^{\pi_1} - v^{\pi_2}||_\infty \le L||\pi_1 - \pi_2||_\infty$, i.e, $v^\pi$ is Lipschitz in $\pi$.

$$
\begin{aligned}
||v^{\pi_1} - v^{\pi_2}||_\infty &= ||T^{\pi_1} v^{\pi_1} - T^{\pi_2} v^{\pi_2}||_\infty \\
&\le ||T^{\pi_1} v^{\pi_1} - T^{\pi_1} v^{\pi_2} + T^{\pi_1} v^{\pi_2} - T^{\pi_2} v^{\pi_2}||_\infty \\
&\le ||T^{\pi_1} v^{\pi_1} - T^{\pi_1} v^{\pi_2}||_\infty + ||T^{\pi_1} v^{\pi_2} - T^{\pi_2} v^{\pi_2}||_\infty \\
&\le \gamma ||v^{\pi_1} - v^{\pi_2}||_\infty + ||T^{\pi_1} v^{\pi_2} - T^{\pi_2} v^{\pi_2}||_\infty, \quad (16)
\end{aligned}
$$

where the last relation is due to the fact $T^{\pi_1}$ is a $\gamma$-contraction. We continue by calculating $|T^{\pi_1} v^{\pi_2} - T^{\pi_2} v^{\pi_2}|(s)$.

$$|T^{\pi_1} v^{\pi_2} - T^{\pi_2} v^{\pi_2}|(s) \le |\sum_a \big(\pi_1(a \mid s) - \pi_2(a \mid s)\big) r(s,a)| + \gamma |\sum_{s'} (P^{\pi_1}_{s',s} - P^{\pi_2}_{s',s}) v^{\pi_2}(s')|. \quad (17)$$

We bound each term in (17). The first term can be bounded by,

$$
\begin{aligned}
|\sum_a \big(\pi_1(a \mid s) - \pi_2(a \mid s)\big) r(s,a)| &\le \sum_a |\big(\pi_1(a \mid s) - \pi_2(a \mid s)\big)||r(s,a)| \\
&\le R_{\max} \max_s \sum_a |(\pi_1(a \mid s) - \pi_2(a \mid s))| \\
&= R_{\max}||\pi_1 - \pi_2||_\infty. \quad (18)
\end{aligned}
$$

In the first relation we used the triangle inequality and in the second inequality the fact that $|r(s,a)|$ is bounded by $R_{\max}$.

The second term in (17) can be bounded by,

$$
\begin{aligned}
|\sum_{s'} (P^{\pi_1}_{s',s} - P^{\pi_2}_{s',s}) v^{\pi_2}(s')| &= |\sum_{s',a} P(s' \mid s,a)(\pi_1(a \mid s) - \pi_2(a \mid s)) v^{\pi_2}(s')| \\
&\le \sum_a \sum_{s'} P(s' \mid s,a)|(\pi_1(a \mid s) - \pi_2(a \mid s)) v^{\pi_2}(s')| \\
&\le \sum_a \sum_{s'} P(s' \mid s,a)|(\pi_1(a \mid s) - \pi_2(a \mid s))||v^{\pi_2}(s')| \\
&\le \sum_a \sum_{s'} P(s' \mid s,a)|(\pi_1(a \mid s) - \pi_2(a \mid s))|\frac{R_{\max}}{1-\gamma} \\
&= \sum_a |(\pi_1(a \mid s) - \pi_2(a \mid s))|\frac{R_{\max}}{1-\gamma} \sum_{s'} P(s' \mid s,a) \\
&= \sum_a |(\pi_1(a \mid s) - \pi_2(a \mid s))|\frac{R_{\max}}{1-\gamma} \\
&\le \max_s \sum_a |(\pi_1(a \mid s) - \pi_2(a \mid s))|\frac{R_{\max}}{1-\gamma} = \frac{R_{\max}}{1-\gamma}||\pi_1 - \pi_2||_\infty \\
& \quad (19)
\end{aligned}
$$

In the first relation we used the triangle inequality, in the forth relation we used the fact that for any $\pi$ and $s$, $v^\pi(s) \le \frac{R_{\max}}{1-\gamma}$, and in the fifth relation the fact that for any $s$ and $a$, $P(s' \mid s,a)$ is a probability function, thus sums to one.

Using (18), (19) to bound (17) yields that for any $s$,

$$|T^{\pi_1} v^{\pi_2} - T^{\pi_2} v^{\pi_2}|(s) \le \frac{R_{\max}}{1-\gamma}||\pi_1 - \pi_2||_\infty.$$

Thus, $||T^{\pi_1}v^{\pi_2} - T^{\pi_2}v^{\pi_2}||_\infty \le \frac{R_{\max}}{1-\gamma}||\pi_1 - \pi_2||_\infty$. Plugging this bound into (16) and rearranging yields,

$$||v^{\pi_1} - v^{\pi_2}||_\infty \le \frac{R_{\max}}{(1-\gamma)^2}||\pi_1 - \pi_2||_\infty, \tag{20}$$

giving that $L = \frac{R_{\max}}{(1-\gamma)^2}$.

We continue by analysing $||T_\kappa v^{\pi_1} - T_\kappa v^{\pi_2}||_\infty$. We remind the reader that $T_\kappa v^\pi$ satisfies the following fixed-point equation:

$$T_\kappa v^\pi(s) = \max_a \left[ r(s,a) + \gamma(1-\kappa)\sum_{s'} P(s' \mid s,a)v^\pi(s') + \kappa\gamma\sum_{s'} P(s' \mid s,a)(T_\kappa v^\pi)(s') \right]$$

$$\overset{\text{def}}{=} \bar{T}_\kappa^\pi T_\kappa v^\pi(s),$$

where we defined the 'optimal' Bellman operator of the surrogate MDP to be $\bar{T}_\kappa^\pi$ (see Remark 1). Furthermore, since this operator is the optimal Bellman operator of a $\kappa\gamma$-discounted MDP, it is a $\kappa\gamma$ contraction mapping. We now use a similar technique as the above to show $||T_\kappa v^{\pi_1} - T_\kappa v^{\pi_2}||_\infty \le L_\kappa ||\pi_1 - \pi_2||_\infty$, i.e, $T_\kappa v^\pi$ is Lipschitz in $\pi$.

$$||T_\kappa v^{\pi_1} - T_\kappa v^{\pi_2}||_\infty = ||\bar{T}_\kappa^{\pi_1} T_\kappa v^{\pi_1} - \bar{T}_\kappa^{\pi_2} T_\kappa v^{\pi_2}||_\infty$$

$$\le ||\bar{T}_\kappa^{\pi_1} T_\kappa v^{\pi_1} - \bar{T}_\kappa^{\pi_1} T_\kappa v^{\pi_2}||_\infty + ||\bar{T}_\kappa^{\pi_1} T_\kappa v^{\pi_2} - \bar{T}_\kappa^{\pi_2} T_\kappa v^{\pi_2}||_\infty$$

$$\le \kappa\gamma||T_\kappa v^{\pi_1} - T_\kappa v^{\pi_2}||_\infty + ||\bar{T}_\kappa^{\pi_1} T_\kappa v^{\pi_2} - \bar{T}_\kappa^{\pi_2} T_\kappa v^{\pi_2}||_\infty.$$

We now bound the second term.

$$|\bar{T}_\kappa^{\pi_1} T_\kappa v^{\pi_2} - \bar{T}_\kappa^{\pi_2} T_\kappa v^{\pi_2}|(s) \le \max_a \gamma(1-\kappa)|\sum_{s'} P(s' \mid s,a)(v^{\pi_1} - v^{\pi_2})(s')|$$

$$\le \max_a \gamma(1-\kappa)\sum_{s'} P(s' \mid s,a)|v^{\pi_1} - v^{\pi_2}|(s')$$

$$\le \max_a \gamma(1-\kappa)\sum_{s'} P(s' \mid s,a)||v^{\pi_1} - v^{\pi_2}||_\infty = \gamma(1-\kappa)||v^{\pi_1} - v^{\pi_2}||_\infty,$$

where we used the definition of $\bar{T}_\kappa^\pi$ and the identity $|\max_{x\in\mathcal{X}} f(x) - \max_{x\in\mathcal{X}} g(x)| \le \max_{x\in\mathcal{X}} |f(x) - g(x)|$ in the first relation and the triangle inequality in the second.

Using (20), we have

$$||T_\kappa v^{\pi_1} - T_\kappa v^{\pi_2}||_\infty \le \frac{\gamma(1-\kappa)}{1-\kappa\gamma}||v^{\pi_1} - v^{\pi_2}||_\infty$$

$$\le \frac{\gamma(1-\kappa)}{1-\kappa\gamma}\frac{R_{\max}}{(1-\gamma)^2}||\pi_1 - \pi_2||_\infty.$$

These results transform to results on $q^\pi$ and $q_\kappa^\pi$ as follows. Starting with $q^\pi$,

$$|q^{\pi_1} - q^{\pi_2}|(s,a) = |r(s,a) + \gamma\sum_{s'} P(s' \mid s,a)v^{\pi_1} - r(s,a) - \gamma\sum_{s'} P(s' \mid s,a)v^{\pi_2}|$$

$$= \gamma|\sum_{s'} P(s' \mid s,a)(v^{\pi_1} - v^{\pi_2})| \le \gamma||v^{\pi_1} - v^{\pi_2}||_\infty.$$

By taking the max-norm on both sides we get the result since $||v^{\pi_1} - v^{\pi_2}||_\infty$ was shown to be Lipschitz in $\pi$.

Next, for $q_\kappa^\pi$ we have

$$|q_\kappa^{\pi_1} - q_\kappa^{\pi_2}|(s,a)$$

$$= |\gamma(1-\kappa)\sum_{s'} P(s' \mid s,a)(v^{\pi_1}(s') - v^{\pi_2}(s')) + \kappa\gamma\sum_{s'} P(s' \mid s,a)(T_\kappa v^{\pi_1} - T_\kappa v^{\pi_2})(s')|$$

$$\le \gamma(1-\kappa)||v^{\pi_1}(s') - v^{\pi_2}(s')||_\infty + \kappa\gamma||T_\kappa v^{\pi_1} - T_\kappa v^{\pi_2}||_\infty.$$

By taking the max-norm on both sides we get the result since, as shown above, both $||v^{\pi_1} - v^{\pi_2}||_\infty$ and $||T_\kappa v^{\pi_1} - T_\kappa v^{\pi_2}||_\infty$ are Lipschitz in $\pi$. Finally, since the vector space is finite (due to the finite state and action space), all $L_p$ norms are equivalent. Thus, the Lipschitzness result applies in any $L_p$ norm as well. □

## C.2 Improvement Step

Here, we prove an equivalent lemma to [16, Lemma 5.4] which shows that the mean value of the process improves. Denote $b_s \equiv b_s(q^\pi, q^\pi_\kappa, \pi)$ as the policy defined in the Algrorithm 1. By using Lemma 7 and setting $\kappa = 0$ we have that

$$v^{(1-\alpha)\pi+\alpha b_s} - v^\pi = \alpha(I - \gamma P^{(1-\alpha)\pi+\alpha b_s})^{-1}(T^{b_s}v^\pi - v^\pi).$$

Thus, by taking the limit $\alpha \to 0$ we have

$$\lim_{\alpha \to 0}(v^{(1-\alpha)\pi+\alpha b_s} - v^\pi) = \alpha \nabla_\pi v^\pi (b_s - \pi)$$

$$= \alpha \langle \nabla_\pi v^\pi, \Delta \pi \rangle$$

$$= \alpha(I - \gamma P^\pi)^{-1}(T^{b_s}v^\pi - v^\pi) + \mathcal{O}(\alpha^2) \geq 0,$$

where the last inequality is since $T^{b_s}v^\pi - v^\pi \geq 0$ by construction and $(I - \gamma P^\pi)^{-1} \geq 0$ componentwise. We thus get that

$$\frac{1}{\alpha}\lim_{\alpha \to 0}(v^{(1-\alpha)\pi+\alpha b_s} - v^\pi) = \langle \nabla_\pi v^\pi, \Delta \pi \rangle \geq 0.$$

## C.3 Convergence of the Algorithm

We define the same Lyapunov function as defined in [16, Lemma 5.5]. Due to previous section it is indeed a Lyapunov function since its derivative is negative and the function is bigger than 0 by construction. The presence of the Lyapunov function leads to the convergence of the policy to the optimal policy, similarly to [16, Corollary 5.6], which leads to the convergence of $q^\pi$ to $q^*$. Lastly, since $T_\kappa v^* = v^*$ [5, Lemma 4] we have that,

$$q^{\pi^*}_\kappa(\pi') = r^{\pi'} + \gamma(1-\kappa)P^{\pi'}v^* + \kappa\gamma P^{\pi'}T_\kappa v^*$$

$$= r^{\pi'} + \gamma(1-\kappa)P^{\pi'}v^* + \kappa\gamma P^{\pi'}v^*$$

$$= r^{\pi'} + \gamma P^{\pi'}v^* = q^*(\pi').$$

which concludes the proof.

# D  Proof of Lemma 4

We first prove a useful lemma that relates the (unnormalized) future distribution, measured in different $\kappa$ scales.

**Lemma 9.** *For any policy $\pi$ and $\kappa, \kappa' \in [0,1]$,*

$$(I - \xi_{\kappa'}D^\pi_{\kappa'}P^\pi)^{-1} = \frac{\kappa'-\kappa}{1-\kappa}I + \frac{1-\kappa'}{1-\kappa}(I - \xi_\kappa D^\pi_\kappa P^\pi)^{-1}.$$

*Proof.* We prove the lemma by using the definition and by some algebraic manipulations.

$$(I - \xi_{\kappa'}D^\pi_{\kappa'}P^\pi)^{-1} = (I - \gamma(1-\kappa')(I - \kappa\gamma'P^\pi)^{-1}P^\pi)^{-1}$$

$$= ((I - \kappa\gamma'P^\pi)^{-1}(I - \kappa\gamma'P^\pi - \gamma(1-\kappa')P^\pi))^{-1}$$

$$= (I - \gamma P^\pi)^{-1}(I - \gamma\kappa'P^\pi)$$

$$= (I - \gamma P^\pi)^{-1} - \kappa'\gamma P^\pi(I - \gamma P^\pi)^{-1}$$

$$= (I - \gamma P^\pi)^{-1} - \kappa'(I + \gamma P^\pi(I - \gamma P^\pi)^{-1} - I)$$

$$= (I - \gamma P^\pi)^{-1} - \kappa'((I - \gamma P^\pi)^{-1} - I)$$

$$= \kappa'I + (1-\kappa')(I - \gamma P^\pi)^{-1}$$

We see that the following relation holds for any $\kappa \in [0,1]$,

$$(I - \gamma P^\pi)^{-1} = \frac{1}{1-\kappa}((I - \xi_\kappa D^\pi_\kappa P^\pi)^{-1} - \kappa I).$$

Plugging this relation into the previous one we get,

$$(I - \xi_{\kappa'} D_{\kappa'}^{\pi} P^{\pi})^{-1} = \kappa' I + (1 - \kappa')(I - \gamma P^{\pi})^{-1}$$
$$= \kappa' I + \frac{1 - \kappa'}{1 - \kappa}((I - \xi_{\kappa} D_{\kappa}^{\pi} P^{\pi})^{-1} - \kappa I)$$
$$= \frac{\kappa' - \kappa}{1 - \kappa} I + \frac{1 - \kappa'}{1 - \kappa}(I - \xi_{\kappa} D_{\kappa}^{\pi} P^{\pi})^{-1}.$$

□

We are now ready to prove Lemma 4. Assume a constant $C_{\kappa}^{\pi^*}(\mu, \nu) < \infty$ such that,

$$d_{\kappa,\mu}^{\pi^*} = (1 - \xi)\mu(I - \xi D_{\kappa}^{\pi^*})^{-1} < C_{\kappa}^{\pi^*}(\mu, \nu)\nu. \tag{21}$$

Given that, we shall calculate $C_{\kappa'}^{\pi^*}(\mu, \nu)$ where $\kappa' > \kappa$.

$$d_{\kappa',\mu}^{\pi^*} = (1 - \xi_{\kappa'})\mu(I - \xi D_{\kappa'}^{\pi^*})^{-1}$$
$$= (1 - \xi_{\kappa'})\left(\frac{\kappa' - \kappa}{1 - \kappa}\mu + \frac{1 - \kappa'}{1 - \kappa}\mu((I - \xi_{\kappa} D_{\kappa}^{\pi} P^{\pi})^{-1})\right)$$
$$\leq \frac{1 - \xi_{\kappa'}}{1 - \kappa}\left((\kappa' - \kappa)\mu + \frac{1 - \kappa'}{1 - \xi_{\kappa}}C_{\kappa}^{\pi^*}(\mu, \nu)\nu\right)$$
$$= \frac{1 - \xi_{\kappa'}}{1 - \kappa}(\kappa' - \kappa + \frac{1 - \kappa'}{1 - \xi_{\kappa}}C_{\kappa}^{\pi^*}(\mu, \nu))(\alpha^* \mu + (1 - \alpha^*)\nu) \overset{\text{def}}{=} C_{\kappa'}^{\pi^*}(\mu, \nu(\alpha))\nu(\alpha),$$

where we used Lemma 9 in the first line, Equation 21 in the second line, and defined $\alpha^* = (1 + \frac{1 - \kappa'}{(1 - \xi_{\kappa})(\kappa' - \kappa)}C_{\kappa}^{\pi^*}(\mu, \nu)))^{-1} \in (0, 1)$ and $C_{\kappa'}^{\pi^*}(\mu, \nu(\alpha^*)) = \frac{1 - \xi_{\kappa'}}{1 - \kappa}(\kappa' - \kappa + \frac{1 - \kappa'}{1 - \xi_{\kappa}}C_{\kappa}^{\pi^*}(\mu, \nu))$. By plugging the expressions of $\xi_{\kappa}, \xi_{\kappa'}$ we see that,

$$C_{\kappa'}^{\pi^*}(\mu, \nu(\alpha^*)) - C_{\kappa}^{\pi^*}(\mu, \nu) = \frac{1 - \xi_{\kappa'}}{1 - \kappa}(\kappa' - \kappa + (\frac{1 - \kappa'}{1 - \xi_{\kappa}} - \frac{1 - \kappa}{1 - \xi_{\kappa'}})C_{\kappa}^{\pi^*}(\mu, \nu))$$
$$= \frac{1 - \xi_{\kappa'}}{1 - \kappa}(\kappa' - \kappa)(1 - C_{\kappa}^{\pi^*}(\mu, \nu)). \tag{22}$$

Since $C_{\kappa}^{\pi^*}(\mu, \nu) \geq 1$ and $\frac{1 - \xi_{\kappa'}}{1 - \kappa}(\kappa' - \kappa) > 0$ we get that $C_{\kappa'}^{\pi^*}(\mu, \nu(\alpha^*)) - C_{\kappa}^{\pi^*}(\mu, \nu) \leq 0$, where the inequality is strict for $C_{\kappa}^{\pi^*}(\mu, \nu) > 1$. Finally, since for $\mu = \nu$ it holds that $\nu(\alpha^*) = (1 - \alpha^*)\nu + \alpha^* \nu = \nu$ for, we get that $C_{\kappa}^{\pi^*}(\nu, \nu)$ is a decreasing function of $\kappa$.

# E    Proof of Theorem 5

We first prove two technical lemmas.

**Lemma 10.** *Let $\pi$ be a policy, $\kappa \in [0, 1]$, $\gamma \in (0, 1)$ and $i \in \mathbb{N} \backslash \{0\}$. Then*

$$(\xi D_{\kappa}^{\pi} P^{\pi})^i = \sum_{t=i-1}^{\infty} \frac{t!}{(i-1)!(t-(i-1))!}\gamma^{t+1}(1 - \kappa)^i \kappa^{t-(i-1)}(P^{\pi})^{t+1},$$

*where, as also given in Definition 4, $D_{\kappa}^{\pi} = (1 - \kappa\gamma)(I - \kappa\gamma P^{\pi})^{-1}$.*

*Proof.* First, for any $x \in \mathbb{R}$ s.t $|x| < 1$ and $i \in \mathbb{N} \backslash \{0\}$ we have that,

$$(1 - x)^{-i} = \sum_{t=i-1}^{\infty} \frac{t!}{(i-1)!(t-(i-1))!}x^{t-(i-1)}.$$

Since it holds that $||\gamma\kappa P^{\pi}|| = \gamma\kappa < 1$, where $|| \cdot ||$ is the spectral norm of the matrix, we can use the same Taylor expansion when replacing $x$ with $\gamma\kappa P^{\pi}$. Thus,

$$(I - \gamma\kappa P^{\pi})^{-i} = \sum_{t=i-1}^{\infty} \frac{t!}{(i-1)!(t-(i-1))!}(\gamma\kappa)^{t-(i-1)}(P^{\pi})^{t-(i-1)}. \tag{23}$$

Since $D_\kappa^\pi = (1 - \kappa\gamma)(I - \kappa\gamma P^\pi)^{-1}$ and any matrix commutes with any function of itself we have that,

$$(\xi D_\kappa^\pi P^\pi)^i = \gamma^i(1-\kappa)^i(D_\kappa^\pi P^\pi)^i = \gamma^i(1-\kappa)^i((I - \kappa\gamma P^\pi)^{-1})^i(P^\pi)^i.$$

By using (23) and packing the terms we conclude the proof.

$$(\xi D_\kappa^\pi P^\pi)^i = \gamma^i(1-\kappa)^i(I - \kappa\gamma P^\pi)^{-i}(P^\pi)^i$$
$$= \sum_{t=i-1}^{\infty} \frac{t!}{(i-1)!(t-(i-1))!}\gamma^{t+1}(1-\kappa)^i\kappa^{t-(i-1)}(P^\pi)^{t+1}$$

$\square$

**Lemma 11.** *Let $\kappa \in [0,1]$, $\gamma \in (0,1), n \in \mathbb{N} \cup \{\infty\}$ and $f : \mathbb{N} \to \mathbb{R}$. Then*

$$\sum_{l=0}^{\infty}\sum_{i=1}^{n-1}\sum_{t=i-1}^{\infty} \frac{t!}{(i-1)!(t-(i-1))!}\gamma^{t+l+1}\kappa^{t-(i-1)}(1-\kappa)^i f(t+1+l)$$
$$\leq (1-\kappa)\sum_{l=0}^{\infty}\sum_{t=0}^{n-2}\gamma^{t+l+1}f(t+1+l) + g(\kappa)(1-\kappa)\kappa\sum_{l=0}^{\infty}\sum_{t=n-1}^{\infty}\gamma^{t+l+1}f(t+1+l),$$

*where $g(\kappa)$ is a bounded function of $\kappa$. When $n \to \infty$ the second term vanishes.*

*Proof.* We start by exchanging the summation indices $i$ and $t$. In order to do so, we decouple the summation to two sums. The range of the indices of the first sum is $t \in \{0,..,n-2\}$ and $i \in \{1,..,t+1\}$ and the range of the indices of the second sum is $t \in \{n-1,..,\infty\}$ and $i \in \{1,..,n-1\}$

$$\sum_{l=0}^{\infty}\sum_{i=1}^{n-1}\sum_{t=i-1}^{\infty} \frac{t!}{(i-1)!(t-(i-1))!}\gamma^{t+l+1}\kappa^{t-(i-1)}(1-\kappa)^i f(t+1+l)$$
$$= \sum_{l=0}^{\infty}\sum_{t=0}^{n-2}\gamma^{t+l+1}f(t+1+l)\sum_{i=1}^{t+1}\frac{t!}{(i-1)!(t-(i-1))!}\kappa^{t-(i-1)}(1-\kappa)^i \quad (24)$$
$$+ \sum_{l=0}^{\infty}\sum_{t=n-1}^{\infty}\gamma^{t+l+1}f(t+1+l)\sum_{i=1}^{n-1}\frac{t!}{(i-1)!(t-(i-1))!}\kappa^{t-(i-1)}(1-\kappa)^i. \quad (25)$$

Let us bound the first sum first (24),

$$\sum_{l=0}^{\infty}\sum_{t=0}^{n-2}\gamma^{t+l+1}f(t+1+l)\sum_{i=1}^{t+1}\frac{t!}{(i-1)!(t-(i-1))!}\kappa^{t-(i-1)}(1-\kappa)^i$$
$$= \sum_{l=0}^{\infty}\sum_{t=0}^{n-2}\gamma^{t+l+1}f(t+1+l)\sum_{i=0}^{t}\frac{t!}{i!(t-i)!}\kappa^{t-i}(1-\kappa)^{i+1}$$
$$= (1-\kappa)\sum_{l=0}^{\infty}\sum_{t=0}^{n-2}\gamma^{t+l+1}f(t+1+l),$$

where in the first line we changed the index summation $i \leftarrow i - 1$ and in the second line we used the binomial identity $\sum_{i=0}^{t}\frac{t!}{i!(t-i)!}\kappa^{t-i}(1-\kappa)^i = (1-\kappa+\kappa)^t = 1$.

In order to bound the second term (25) we define the following function, $\tilde{g} : [n-1,\infty) \to \mathbb{R}$,

$$\tilde{g}(t) \stackrel{\text{def}}{=} \sum_{i=0}^{n-2}\frac{t!}{i!(t-i)!}\kappa^{t-i}(1-\kappa)^i.$$

The function $\tilde{g}(t)$ is a sum of polynomial terms multiplied by a geometric decaying term, $\kappa^t$. Thus, this function is bounded from above, i.e, exists $t^* \in [n-1,\infty)$ such that $\tilde{g}(t) \leq \tilde{g}(t^*)$, $\forall t \in [n-1,\infty)$.

For such $t^*$, by construction, we have that

$$\sum_{i=1}^{n-1} \frac{t!}{(i-1)!(t-(i-1))!} \kappa^{t-(i-1)}(1-\kappa)^i = (1-\kappa) \sum_{i=0}^{n-2} \frac{t!}{i!(t-i)!} \kappa^{t-i}(1-\kappa)^i$$

$$\leq (1-\kappa) \sum_{i=0}^{n-2} \frac{t^*!}{i!(t^*-i)!} \kappa^{t^*-i}(1-\kappa)^i$$

$$= (1-\kappa)\kappa^{t^*-(n-2)} \sum_{i=0}^{n-2} \frac{t^*!}{i!(t^*-i)!} \kappa^{(n-2)-i}(1-\kappa)^i$$

$$\leq (1-\kappa)\kappa \sum_{i=0}^{n-2} \frac{t^*!}{i!(t^*-i)!} \kappa^{(n-2)-i}(1-\kappa)^i$$

where the last line holds since for $\kappa \in [0,1]$, $t^* \in [n-1,\infty)$ it holds that $\kappa^{t^*-(n-2)} \leq \kappa$. We now define $g(\kappa) \stackrel{\text{def}}{=} \sum_{i=0}^{n-2} \frac{t^*!}{i!(t^*-i)!} \kappa^{(n-2)-i}(1-\kappa)^i$, and observe that it is a bounded function of $\kappa \in [0,1]$, since it is a sum of positive powers of $\kappa$. Thus, (25) is bounded by

$$\sum_{l=0}^{\infty} \sum_{t=n-1}^{\infty} \gamma^{t+l+1} f(t+1+l) \sum_{i=1}^{n-1} \frac{t!}{(i-1)!(t-(i-1))!} \kappa^{t-(i-1)}(1-\kappa)^i$$

$$\leq g(\kappa)(1-\kappa)\kappa \sum_{l=0}^{\infty} \sum_{t=n-1}^{\infty} \gamma^{t+l+1} f(t+1+l)$$

Finally, for the case $n=\infty$ observe we can repeat the same analysis we did for the first term (24) without the need to decouple to two sums. Thus, for this case, the bound on the first term, with $n=\infty$, bounds the expression.

$\square$

We are now ready to prove Theorem 5. The proof strategy is similar to the line of work in [7, 19, 11]: Keeping track of the cumulative error and using the definition of $c(i)$ and $c^{\pi^*}(i)$, we bound the performance loss in the $\mu$-weighted $L_1$ norm.

Since the policy in each iteration is an approximate $\kappa$-greedy policy (see Definition 2), it holds that $\nu T_\kappa^{\pi_k} v^{\pi_{k-1}} \geq \nu T_\kappa v^{\pi_{k-1}} - \delta$ in each iteration. Let the error vector at the $i$-th iteration $\bar{\delta}_i$ satisfy $\nu \bar{\delta}_i \leq \delta$. Thus,

$$v^* - v^{\pi_k} = T_\kappa^{\pi^*} v^* - T_\kappa^{\pi^*} v^{\pi_{k-1}} + T_\kappa^{\pi^*} v^{\pi_{k-1}} - v^{\pi_k}$$

$$= \xi D_\kappa^{\pi^*} P^{\pi^*}(v^* - v^{\pi_{k-1}}) + T_\kappa^{\pi^*} v^{\pi_{k-1}} - v^{\pi_k}$$

$$= \xi D_\kappa^{\pi^*} P^{\pi^*}(v^* - v^{\pi_{k-1}}) + T_\kappa^{\pi^*} v^{\pi_{k-1}} - T_\kappa^{\pi_k} v^{\pi_{k-1}} + T_\kappa^{\pi_k} v^{\pi_{k-1}} - v^{\pi_k}$$

$$\leq \xi D_\kappa^{\pi^*} P^{\pi^*}(v^* - v^{\pi_{k-1}}) + T_\kappa^{\pi^*} v^{\pi_{k-1}} - \max_{\pi'} T_\kappa^{\pi'} v^{\pi_{k-1}} + \bar{\delta}_i + T_\kappa^{\pi_k} v^{\pi_{k-1}} - v^{\pi_k}$$

$$\leq \xi D_\kappa^{\pi^*} P^{\pi^*}(v^* - v^{\pi_{k-1}}) + \bar{\delta}_i + T_\kappa^{\pi_k} v^{\pi_{k-1}} - v^{\pi_k}$$

$$= \xi D_\kappa^{\pi^*} P^{\pi^*}(v^* - v^{\pi_{k-1}}) + \bar{\delta}_i + \xi D_\kappa^{\pi_\kappa} P^{\pi_k}(v^{\pi_{k-1}} - v^{\pi_k}), \tag{26}$$

where we used in the second and last relations that for any policy $\pi$, and any value functions $v_1, v_2$, $T_\kappa^\pi v_1 - T_\kappa^\pi v_2 = \xi D_\kappa^\pi P^\pi(v_1 - v_2)$. This can be seen by using the definition of $T_\kappa^\pi$ (see Section 3). Notice that

$$v^{\pi_{k-1}} - v^{\pi_k} = T_\kappa^{\pi_{k-1}} v^{\pi_{k-1}} - v^{\pi_k}$$

$$\leq \max_{\pi'} T_\kappa^{\pi'} v^{\pi_{k-1}} - v^{\pi_k}$$

$$\leq T_\kappa^{\pi_k} v^{\pi_{k-1}} - v^{\pi_k} + \bar{\delta}_i$$

$$= T_\kappa^{\pi_k} v^{\pi_{k-1}} - T_\kappa^{\pi_k} v^{\pi_k} + \bar{\delta}_i$$

$$= \xi D_\kappa^{\pi_k} P^{\pi_k}(v^{\pi_{k-1}} - v^{\pi_k}) + \bar{\delta}_i.$$

Hence,

$$(I - \xi D_\kappa^{\pi_k} P^{\pi_k})(v^{\pi_{k-1}} - v^{\pi_k}) \le \bar{\delta}_i, \text{ i.e.,}$$

$$v^{\pi_{k-1}} - v^{\pi_k} \le (I - \xi D_\kappa^{\pi_k} P^{\pi_k})^{-1}\bar{\delta}_i. \tag{27}$$

The last equation holds due to [14, Lemma 4.2], combined with the fact that $(I - \xi D_\kappa^{\pi_k} P^{\pi_k})^{-1} = \sum_{i=0}^\infty \xi D_\kappa^{\pi_k} P^{\pi_k} \ge 0$, element-wise.

Plugging (27) into (26), we have that

$$v^* - v^{\pi_k} \le \xi D_\kappa^{\pi^*} P^{\pi^*}(v^* - v^{\pi_{k-1}}) + \bar{\delta}_i + \xi D_\kappa^{\pi_k} P^{\pi_k}(I - \xi D_\kappa^{\pi_k} P^{\pi_k})^{-1}\bar{\delta}_i$$

$$= \xi D_\kappa^{\pi^*} P^{\pi^*}(v^* - v^{\pi_{k-1}}) + (I - \xi D_\kappa^{\pi_k} P^{\pi_k})^{-1}\bar{\delta}_i,$$

where the second relation holds since for matrix $X$ s.t. $\|X\| < 1$, $I + X(I - X)^{-1} = (I - X)^{-1}$.
We thus get that the errors accumulate as follows.

$$v^* - v^{\pi_k} \le \sum_{i=0}^{k-1}(\xi D_\kappa^{\pi^*} P^{\pi^*})^i(I - \xi D_\kappa^{\pi_{k-i}} P^{\pi_{k-i}})^{-1}\bar{\delta}_i + (\xi D_\kappa^{\pi^*} P^{\pi^*})^k(v^* - v^{\pi_0}).$$

We continue by multiplying both sides with $\mu$ and get

$$\mu(v^* - v^{\pi_k}) \le \sum_{i=0}^{k-1}\mu(\xi D_\kappa^{\pi^*} P^{\pi^*})^i(I - \xi D_\kappa^{\pi_{k-i}} P^{\pi_{k-i}})^{-1}\bar{\delta}_i + \xi^k \frac{R_{\max}}{1-\gamma}. \tag{28}$$

Using Lemma 9 with $\kappa = 0$ and renaming $\kappa'$ to $\kappa$, we have that

$$(I - \xi D_\kappa^{\pi_{k-i}} P^{\pi_{k-i}})^{-1} = (1 - \kappa)(I - \gamma P^{\pi_{k-i}})^{-1} + \kappa I.$$

Plugging this relation into (28) gives

$$\mu(v^* - v^{\pi_k}) \le \sum_{i=0}^{k-1}\mu(\xi D_\kappa^{\pi^*} P^{\pi^*})^i((1 - \kappa)(I - \gamma P^{\pi_{k-i}})^{-1} + \kappa I)\bar{\delta}_i + \xi^k \frac{R_{\max}}{1-\gamma}$$

$$\le (1 - \kappa)\sum_{i=0}^{k-1}\mu(\xi D_\kappa^{\pi^*} P^{\pi^*})^i(I - \gamma P^{\pi_{k-i}})^{-1}\bar{\delta}_i + \kappa\sum_{i=0}^{k-1}\mu(\xi D_\kappa^{\pi^*} P^{\pi^*})^i\bar{\delta}_i + \xi^k \frac{R_{\max}}{1-\gamma}. \tag{29}$$

The following two lemmas provide bounds for the first two terms above. The bounds are composed of the concentrability coefficients (see Definition 3 and Definition 4).

**Lemma 12.** *Let $\kappa \in [0, 1]$. For any sequence of policies $\{\pi_{k-i}\}_{i=0}^{k-1}$, optimal policy $\pi^*$, and error vector which satisfy $\nu\bar{\delta}_i \le \delta$,*

$$\sum_{i=0}^{k-1}\mu(\xi D_\kappa^{\pi^*} P^{\pi^*})^i(I - \gamma P^{\pi_{k-i}})^{-1}\bar{\delta}_i \le \left(\frac{(1-\kappa)C^{(2)}(\mu,\nu)}{(1-\gamma)^2} + \frac{\kappa C^{(1)}(\mu,\nu)}{1-\gamma}\right)\delta \tag{30}$$

*and*

$$\sum_{i=0}^{k-1}\mu(\xi D_\kappa^{\pi^*} P^{\pi^*})^i(I - \gamma P^{\pi_{k-i}})^{-1}\bar{\delta}_i$$

$$\le \left(k\frac{(1-\kappa)C^{(1)}(\mu,\nu)}{1-\gamma} + \frac{\kappa C^{(1)}(\mu,\nu)}{1-\gamma} + \frac{g(\kappa)(1-\kappa)\kappa\gamma^k C^{(2,k)}(\mu,\nu)}{(1-\gamma)^2}\right)\delta. \tag{31}$$

*Proof.* We start with proving (30). Let $\pi'$ be an arbitrary policy. For $i > k - 1$, we define $\pi_{k-i} = \pi'$ and vectors $\bar{\delta}_i$ s.t. $\nu\bar{\delta}_i \le \delta$ .

$$\sum_{i=0}^{k-1} \mu(\xi D_\kappa^{\pi^*} P^{\pi^*})^i (I - \gamma P^{\pi_{k-i}})^{-1}\bar{\delta}_i \le \sum_{i=0}^{\infty} \mu(\xi D_\kappa^{\pi^*} P^{\pi^*})^i (I - \gamma P^{\pi_{k-i}})^{-1}\bar{\delta}_i$$

$$= \mu(I - \gamma P^{\pi_k})^{-1}\bar{\delta}_0 + \sum_{i=1}^{\infty} \mu(\xi D_\kappa^{\pi^*} P^{\pi^*})^i (I - \gamma P^{\pi_{k-i}})^{-1}\bar{\delta}_i. \tag{32}$$

For the first term in (32) we have that

$$\mu(I - \gamma P^{\pi_k})^{-1}\bar{\delta}_0 = \sum_{l=0}^{\infty} \gamma^l \mu(P^{\pi_k})^l\bar{\delta}_0 \le \sum_{l=0}^{\infty} \gamma^l c(l)\nu\bar{\delta}_0 \le \sum_{l=0}^{\infty} \gamma^l c(l)\delta = \frac{C^{(1)}(\mu,\nu)}{1-\gamma}\delta, \tag{33}$$

where for the second relation we used the definition of the sequence $\{c(i)\}_{i=0}^{\infty}$ (see Definition 3) and in the third relation we used $\nu\bar{\delta}_0 \le \delta$ (see Definition 2).

Next, we bound the second term in (32).

$$\sum_{i=1}^{\infty} \mu(\xi D_\kappa^{\pi^*} P^{\pi^*})^i (I - \gamma P^{\pi_{k-i}})^{-1}\bar{\delta}_i$$

$$= \sum_{l=0}^{\infty}\sum_{i=1}^{\infty} \gamma^l \mu(\xi D_\kappa^{\pi^*} P^{\pi^*})^i (P^{\pi_{k-i}})^l\bar{\delta}_i \tag{34}$$

$$= \sum_{l=0}^{\infty}\sum_{i=1}^{\infty}\sum_{t=i-1}^{\infty} \frac{t!}{(i-1)!(t-(i-1))!}\gamma^{l+t+1}\kappa^{t-(i-1)}(1-\kappa)^i\mu(P^{\pi^*})^{t+1}(P^{\pi_{k-i}})^l\bar{\delta}_i$$

$$\le \sum_{l=0}^{\infty}\sum_{i=1}^{\infty}\sum_{t=i-1}^{\infty} \frac{t!}{(i-1)!(t-(i-1))!}\gamma^{l+t+1}\kappa^{t-(i-1)}(1-\kappa)^i c(t+1+l)\delta$$

$$\le (1-\kappa)\sum_{l=0}^{\infty}\sum_{t=0}^{\infty} \gamma^{l+t+1}c(t+1+l)\delta \tag{35}$$

$$= (1-\kappa)\sum_{l=0}^{\infty}\sum_{t=1}^{\infty} \gamma^{l+t}c(t+l)\delta$$

$$= (1-\kappa)\left(\sum_{l=0}^{\infty}\sum_{t=1}^{\infty} \gamma^{l+t}c(t+l) + \sum_{l=0}^{\infty}\gamma^l c(l) - \sum_{l=0}^{\infty}\gamma^l c(l)\right)\delta$$

$$= (1-\kappa)\left(\sum_{l=0}^{\infty}\sum_{t=0}^{\infty} \gamma^{l+t}c(t+l) - \sum_{l=0}^{\infty}\gamma^l c(l)\right)\delta = (1-\kappa)\left(\frac{C^{(2)}(\mu,\nu)}{(1-\gamma)^2} - \frac{C^{(1)}(\mu,\nu)}{1-\gamma}\right)\delta. \tag{36}$$

For the first relation we used the Taylor expansion $(I - \gamma P^{\pi_{k-i}})^{-1} = \sum_{l=0}^{\infty}\gamma^l (P^{\pi_{k-i}})^l$, for the second we used Lemma 10, for the third we used the definition of the sequence $\{c(i)\}_{i=0}^{\infty}$ and $\nu\bar{\delta}_i \le \delta$, for the fourth we applied Lemma 11 with $n = \infty$ and $f(\cdot) = c(\cdot)$, and for the fifth we shifted the summation index $t \leftarrow t + 1$.

We bound (32) by summing the bounds in (33) and (36) to obtain the first statement of the lemma, (30).

To prove the second statement, (31), we again split expression of interest, similarly to (32).

$$\sum_{i=0}^{k-1} \mu(\xi D_\kappa^{\pi^*} P^{\pi^*})^i (I - \gamma P^{\pi_{k-i}})^{-1}\bar{\delta}_i \le \mu(I - \gamma P^{\pi_k})^{-1}\bar{\delta}_0 + \sum_{i=1}^{k-1} \mu(\xi D_\kappa^{\pi^*} P^{\pi^*})^i (I - \gamma P^{\pi_{k-i}})^{-1}\bar{\delta}_i. \tag{37}$$

As in (33), the first term in (37) is bounded by

$$\mu(I - \gamma P^{\pi_k})^{-1}\bar{\delta}_0 \le \frac{C^{(1)}(\mu,\nu)}{1-\gamma}\delta. \tag{38}$$

Next, we bound the second term in (37).

$$\sum_{i=1}^{k-1}\mu(\xi D_\kappa^{\pi^*}P^{\pi^*})^i(I-\gamma P^{\pi_{k-i}})^{-1}\bar\delta_i$$

$$=\sum_{l=0}^{\infty}\sum_{i=1}^{k-1}\gamma^l\mu(\xi D_\kappa^{\pi^*}P^{\pi^*})^i(P^{\pi_{k-i}})^l\bar\delta_i$$

$$\leq(1-\kappa)\sum_{l=0}^{\infty}\sum_{t=0}^{k-2}\gamma^{t+1+l}c(t+1+l)\delta+g(\kappa)(1-\kappa)\kappa\sum_{l=0}^{\infty}\sum_{t=k-1}^{\infty}\gamma^{t+1+l}c(t+1+l)\delta$$

$$=(1-\kappa)\sum_{t=0}^{k-2}\sum_{l=0}^{\infty}\gamma^{t+1+l}c(t+1+l)\delta+g(\kappa)(1-\kappa)\kappa\gamma^k\sum_{l=0}^{\infty}\sum_{t=0}^{\infty}\gamma^{t+l}c(t+l+k)\delta$$

$$\leq(k-1)\frac{(1-\kappa)C^{(1)}(\mu,\nu)}{1-\gamma}\delta+\frac{g(\kappa)(1-\kappa)\kappa\gamma^k C^{(2,k)}(\mu,\nu)}{(1-\gamma)^2}\delta. \tag{39}$$

In the first relation we used the Taylor expansion of $(I-\gamma P^{\pi_{k-i}})$. For the second relation we perform the same steps as from (34) to (35), where this time we used Lemma 11 with finite $n=k$.

Summing the terms in (38) and (39), we obtain the second statement of the lemma, (31). $\qquad\square$

**Lemma 13.** *Let $\kappa\in[0,1]$. For any sequence of policies $\{\pi_{k-i}\}_{i=0}^{k-1}$, optimal policy $\pi^*$, and error vectors which satisfy $\nu\bar\delta_i\leq\delta$, ,*

$$\sum_{i=0}^{k-1}\mu(\xi D_\kappa^{\pi^*}P^{\pi^*})^i\bar\delta_i\leq\frac{1-\kappa\gamma}{1-\gamma}C_\kappa^{\pi^*(1)}(\mu,\nu)\delta \tag{40}$$

*and*

$$\sum_{i=0}^{k-1}\mu(\xi D_\kappa^{\pi^*}P^{\pi^*})^i\bar\delta_i\leq k\frac{1-\kappa\gamma}{1-\gamma}C_\kappa^{\pi^*}(\mu,\nu)\delta. \tag{41}$$

*Proof.* We begin proving the first statement. For $i>k-1$, we define vectors $\bar\delta_i$ s.t. $\nu\bar\delta_i\leq\delta$. Thus,

$$\sum_{i=0}^{k-1}\mu(\xi D_\kappa^{\pi^*}P^{\pi^*})^i\bar\delta_i\leq\mu\bar\delta_0+\sum_{i=1}^{\infty}\mu(\xi D_\kappa^{\pi^*}P^{\pi^*})^i\bar\delta_i. \tag{42}$$

For the first term in (42),

$$\mu\bar\delta_0\leq c(0)\nu\bar\delta_0\leq c(0)\delta, \tag{43}$$

where we used Definition 3 and then Definition 2.

For the second term in (42), we have

$$\sum_{i=1}^{\infty}\mu(\xi D_\kappa^{\pi^*}P^{\pi^*})^i\bar\delta_i$$

$$=\sum_{i=1}^{\infty}\sum_{t=i-1}^{\infty}\frac{t!}{(i-1)!(t-(i-1))!}\gamma^{t+1}(1-\kappa)^i\kappa^{t-(i-1)}\mu(P^{\pi^*})^{t+1}\bar\delta_i$$

$$\leq\sum_{i=1}^{\infty}\sum_{t=i-1}^{\infty}\frac{t!}{(i-1)!(t-(i-1))!}\gamma^{t+1}(1-\kappa)^i\kappa^{t-(i-1)}c^{\pi^*}(t+1)\delta$$

$$\leq(1-\kappa)\sum_{t=0}^{\infty}\gamma^{t+1}c^{\pi*}(t+1)\delta$$

$$=(1-\kappa)\sum_{t=0}^{\infty}\gamma^t c^{\pi*}(t)\delta-(1-\kappa)c(0)\delta=\frac{(1-\kappa)C^{\pi^*(1)}(\mu,\nu)}{1-\gamma}\delta-(1-\kappa)c(0)\delta. \tag{44}$$

For the first relation we apply Lemma 10, for the second we use the definition of $\{c^{\pi^*}(i)\}_{i=0}^{\infty}$ and use $\nu \bar{\delta}_i \leq \delta$. For the third relation we apply Lemma 11 with $n = \infty$, $f(\cdot) = c^{\pi^*}(\cdot)$ and drop the $l$ summation.

Summing the terms in (43) and (44), we get

$$\sum_{i=0}^{k-1} \mu(\xi D_\kappa^{\pi^*} P^{\pi^*})^i \bar{\delta}_i \leq \frac{1}{1-\gamma} \left( (1-\kappa) C^{\pi^*(1)}(\mu, \nu) + (1-\gamma)\kappa c(0) \right) \delta = \frac{1-\kappa\gamma}{1-\gamma} C_\kappa^{\pi^*(1)}(\mu, \nu)\delta,$$

where we identify $C_\kappa^{\pi^*(1)}(\mu, \nu)$ to be the same expression as in Definition 4.

For the second statement of the lemma, (41), we use the identity $(\xi D_\kappa^{\pi^*} P^{\pi^*})^i \leq (I - \xi D_\kappa^{\pi^*} P^{\pi^*})^{-1}$:

$$\sum_{i=0}^{k-1} \mu(\xi D_\kappa^{\pi^*} P^{\pi^*})^i \bar{\delta}_i \leq \sum_{i=0}^{k-1} \mu(I - \xi D_\kappa^{\pi^*} P^{\pi^*})^{-1} \bar{\delta}_i$$

$$\leq \sum_{i=0}^{k-1} \frac{C_\kappa^{\pi^*}(\mu, \nu)}{1-\xi} \nu \bar{\delta}_i \leq k \frac{C_\kappa^{\pi^*}(\mu, \nu)}{1-\xi} \delta = k \frac{1-\kappa\gamma}{1-\gamma} C_\kappa^{\pi^*}(\mu, \nu)\delta,$$

where the second relation holds due to the definition of $C_\kappa^{\pi^*}(\mu, \nu)$. $\qquad\square$

So far, the proof went as follows. First, we expressed the cumulative error in (29) as the sum of three terms. Bounding the first and second terms is done with Lemmas 12 and 13, respectively. Each of those two lemmas gives bounds of two forms. These two forms correspond to the two statements in Theorem 5. We now apply the bounds so as to obtain the first statement. Specifically, plugging (30) and (40) into (29) gives the first statement in Theorem 5.

To obtain the second statement of Theorem 5, we apply the second form of the bounds in Lemmas 12 and 13. Specifically, we plug (31) and (41) into (29). This gives

$$\mu(v^* - v^{\pi_k})$$
$$\leq \left( k\frac{\kappa C_\kappa^{\pi^*}(\mu, \nu)}{1-\xi} + k\frac{(1-\kappa)^2 C^{(1)}(\mu, \nu)}{1-\gamma} + \frac{(1-\kappa)\kappa C^{(1)}(\mu, \nu)}{1-\gamma} + \frac{g(\kappa)(1-\kappa)^2 \kappa \gamma^k C^{(2,k)}(\mu, \nu)}{(1-\gamma)^2} \right) \delta$$
$$+ \xi^k \frac{R_{\max}}{1-\gamma}.$$

We now carefully choose the iteration number $k$ to make the last term smaller than $\delta$:

$$k^* = \left\lceil \frac{\log \frac{R_{max}}{\delta(1-\gamma)}}{1-\xi} \right\rceil = \left\lceil \frac{(1-\kappa\gamma) \log \frac{R_{max}}{\delta(1-\gamma)}}{1-\gamma} \right\rceil. \tag{45}$$

By doing so we see that $\xi^{k^*} \frac{R_{\max}}{1-\gamma} < \delta$ and obtain the second statement of the result.

# F   Proof of Theorem 6

Here, we merely follow the arguments of [19, Appendix A], while using the operators $T_\kappa^\pi$ instead of $T^\pi$ and the approximate operator defined in Definition 2. As in Section E, we define the component-wise error at the $i$-th iteration, $\bar{\delta}_i$, which satisfies $\nu \bar{\delta}_i \leq \delta$. We have that for all $k$,

$$v^* - v^{\sigma_{\kappa,k}} = T_\kappa^{\pi^*} v^* - T_\kappa^{\pi^*} v^{\sigma_{k-1}} + T_\kappa^{\pi^*} v^{\sigma_{k-1}} - T_\kappa^{\pi_k} v^{\sigma_{k-1}}$$
$$\leq \xi D_\kappa^{\pi^*} P^{\pi^*} (v^* - v^{\sigma_{k-1}}) + \bar{\delta}_k.$$

Thus, by induction on $k$, we obtain:

$$v^* - v^{\sigma_{\kappa,k}} \leq \sum_{i=0}^{k-1} (\xi D_\kappa^{\pi^*} P^{\pi^*})^i \bar{\delta}_i + (\xi D_\kappa^{\pi^*} P^{\pi^*})^k (v^* - v^{\pi_0})$$

$$\leq \sum_{i=0}^{k-1} (\xi D_\kappa^{\pi^*} P^{\pi^*})^i \bar{\delta}_i + \xi^k \frac{R_{\max}}{1-\gamma}$$

We can directly bound this term by applying Lemma 13. The two statements in that lemma lead to the two statements in Theorem 6. Again, for the second statement, we set $k$ as in (45).