[Reviews · NeurIPS 2018]

Reviewer 1



This paper provides the analysis of T-greedy Policy Iteration methods, both in the online and approximate settings. In this context the paper makes several points. First, it shows that multiple-step greedy policies using soft, step-sized updates is not guaranteed to make improvements. Second, to address the above issue, the authors propose a "cautious" improvement operator that works at two time scales: first it solves the optimal policy of small-horizon MDP with shaped reward, and then it uses the computed value to shape the reward for the next iteration. Third, the paper quantifies the tradeoff between short-horizon bootstrap bias and long-rollout variance, which corresponds to the classic \lambda tradeoff in TD(\lambda). This is a dense, theoretical paper. While the authors try to provide intuition behind their results, I still found the paper rather difficult to follow. Ideally, the paper would provide some experimental results to aid the intuition, but it doesn't. This is a missed opportunity in my opinon. There are many questions such expriments could answer. What is the impact of "cautious" updates on convergence? Quantitatively quantify the tradeoff between short-horizon bootstrap bias and long-rollout variance, and many more. Otherwise, the results presented in the paper are significant. In particular, I can see the "cautious" updates having a considerable impact in practice, assuming experimental results are going to convincingly illustarte that there are cases of practical interest in which just using soft, step-sized updates fails to converge. As a final note, I was not able to double-check the proofs so please take this review with a (big) grain of salt.

Reviewer 2



Summary: this paper proposed some online and approximate PI algorithms. The authors also provide a thorough analysis for the proposed algorithms. The proposed algorithm generalised a previous work 1-step-greedy policy. The analysis is thorough to understand the properties and guaranteed performance of k-greedy policy. The authors found out that the proposed method k-greedy policy does not guarantee to perform better than 1-step policy. From this finding, the paper proposes ways to circumvent this and thus obtain some non-trivial performance guarantees. Overall: In general, the paper is very well written. The ideas are solid with detailed analysis from which circumvent ideas can be proposed. Though analysis is based on many previous work, the results are significant. Thought the algorithm and analysis are theoretically sound, it would be nicer if the paper has a small set of proof-concept experiments like ones in [5]. - Section 5: Does the generative model G(\nu,\pi) have any specific dependencies on \nu and \pi? Or is it just the underlying environment for a RL problem? - Analysis is based on an assumption of discrete problems. I wonder how the idea and analysis can be extended to the case of large or continuous problem. Could the authors make any comments on this? * Minor comments: - Line 256 page 8: should it be (1-\xi) at denominator of the term after ".r.. is that ..." - In Algorithm 2 and 3, the notation for G_{\kappa,\delta,\nu} is inconsistent.

Reviewer 3



Summary: The paper studies multi-step greedy policies and their use in approximate policy iteration. The authors first show a negative result that soft-policy updates using the multi-step greedy policies do not guarantee policy improvement. Then the authors proposed an algorithm that uses cautious soft updates (only update to the kappa greedy policy only when assured to improve, otherwise stay with one-step greedy policy) and show that it converges to the optimal policy. Lastly the authors studied hard updates by extending APIs to multi-step greedy policy setting. Comments: 1. Theorem 2 presents an interesting and surprising result. Though the authors presented the example in the proof sketch, but I wonder if the authors could provide more intuitions behind this? Based on the theorem, for multi-step greedy policy, it seems that h needs to be bigger than 2. So I suspect that h = 2 will still work (meaning there could exist small alpha)? Obviously h = 1 works, but then why when h = 3, the soft-update suddenly stops working unless alpha is exactly equal to 1? I would expect that one would require larger alpha when h gets larger. 2. Alg 1 and theorem 4 are nice. While I didn’t check the proof of theorem 4, how fast the algorithm will converge compare to a simple one-step greedy version? Namely what is the exact benefit of introducing the extra q_{\kappa} here (lemma 3 shows that the contraction is still gamma, and has nothing to do with kappa here)? 3. Regarding the hard updates, how realistic is the assumption on the existence of a kappa-greedy policy oracle? This is different from a greedy oracle that one would typically use in, for instance, CPI, where the oracle can be implemented by classic cost-sensitive classification with cost-to-go evaluated simply by roll-outs. To compute such a kappa-greedy policy here, we essentially need to solve a MDP (i.e. planning), though it is reshaped and the discount factor is decreased. Using a conservative policy iteration for such a greedy oracle is strange. Why not just using CPI to solve the original problem? Can we show that using CPI as the oracle for computing kappa-greedy policy eventually resulting a faster algorithm than CPI itself, if one set kappa optimally? Similarly, can we show that using CPI as the kappa greedy policy oracle in PSDP resulting a faster algorithm than PSDP? After rebuttal: Thanks for detailed comments in the rebuttal. Though understanding the limitation of kappa-greedy oracle and how to set kappa are important for practical applications of the proposed approaches, I do think the paper already contains significant contribution. I've updated my score.